# SAFE: A Novel Approach to AI Weather Evaluation through Stratified Assessments of Forecasts over Earth

## Abstract

The dominant paradigm in machine learning is to assess model performance based on average loss across all samples in some test set. This amounts to averaging performance geospatially across the Earth in weather and climate settings, failing to account for the non-uniform distribution of human development and geography. We introduce Stratified Assessments of Forecasts over Earth (SAFE), a package for elucidating the stratified performance of a set of predictions made over Earth. SAFE integrates various data domains to stratify by different attributes associated with geospatial gridpoints: territory (usually country), global subregion, income, and landcover (land or water). This allows us to examine the performance of models for each individual stratum of the different attributes (e.g., the accuracy in every individual country). To demonstrate its importance, we utilize SAFE to benchmark a zoo of state-of-the-art AI-based weather prediction models, finding that they all exhibit disparities in forecasting skill across every attribute. We use this to seed a benchmark of model forecast fairness through stratification at different lead times for various climatic variables. By moving beyond globally-averaged metrics, we for the first time ask: where do models perform best or worst, and which models are most fair? To support further work in this direction, the SAFE package is open source and available at `https://anonymous.4open.science/r/safe-E7C7`.

## 1 Introduction

Artificial intelligence weather prediction (AIWP) models, alternatively machine learning weather prediction (MLWP) models or neural weather models (NWM), are becoming increasingly competitive with traditional numerical weather prediction (NWP) models. All of these approaches are typically used in making medium-range weather forecasts (interchangeably, "prediction"). The range of a forecast is determined by its lead time $\tau$. When a weather prediction model is fed the state of variables at time $d$, its task is to predict the state of those variables (or some subset of them) at time $d+\tau$. There is no consistent definition for medium-range, with the European Centre for Medium-Range Weather Forecasts (ECMWF) defining it as any prediction made with $\tau$ (or $n \times \tau$ if taking an autoregressive rollout of $n$ steps) within 0–15 days (European Centre for Medium-Range Weather Forecasts, 2025b), while other sources more narrowly define it as 3–7 days (Meteorological Society; Weather Prediction Center). AIWP models are seeing increasing adoption in interfaces where they provide these medium-range forecasts, from Google's Weather app (Leffer, 2024) to various experimental models at the National Oceanic and Atmospheric Administration (NOAA) (Potvin et al., 2025; Sadeghi Tabas et al., 2025).

Root mean square error (RMSE) is the preeminent metric used in assessing the quality of AIWP models (Radford et al., 2025; Rasp et al., 2020). The general form of RMSE is

$$\sqrt{\frac{\Sigma_{y \in Y}(\hat{y} - y)^2}{|Y|}} \tag{1}$$

where $Y$ is the set of all ground truth variable values that a model is trying to predict, and $\hat{y}$ is the model's prediction for each corresponding $y \in Y$. Every $y$ is the value of some variable (e.g., temperature) at some point in time $d \in D$, longitude $i \in I$, latitude $j \in J$, and, for certain atmospheric variables, vertical level $v \in V$.

There are various approaches for how different models handle being able to make predictions at different lead times. The naive approach is to train a model with the ability to predict some fixed $\tau' \in T$ amount of time in the future, where $T$ is a set of durations. This allows the model to forecast the weather with temporal resolution of $\tau'$ (i.e., multiples of $\tau'$ after the timestamp of the input variables) through autoregressive rollout. This is the approach taken by Keisler (Keisler, 2022), the Spherical CNN (Esteves et al., 2023), FourCastNet (Pathak et al., 2022), and the spherical Fourier neural operator (SFNO) (Bonev et al., 2023), all with $\tau' = 6$ hours. Pangu-Weather (Bi et al., 2022) trains four different models, each with a different, fixed lead time. This is used in tandem with a greedy algorithm that minimizes the number of autoregressive steps that need to be taken to make a prediction at any given lead time (which must be a multiple of their smallest lead time model). FuXi (Chen et al., 2023) uses a cascaded set of three different models that cover different ranges of lead times.

The square of RMSE, mean squared error (MSE), frequently referred to as the $L2$ loss, is often used as a training objective. This is the case for Spherical CNN (Esteves et al., 2023) and GenNet (Lopez-Gomez et al., 2023). GraphCast (Lam et al., 2023) and GenCast (Price et al., 2023) use weighted MSE loss functions. Keisler takes a weighted sum of MSE values (Keisler, 2022). NeuralGCM (Kochkov et al., 2024) has a five-term loss function, each of which is a variation of MSE. FuXi (Chen et al., 2023) uses the mean absolute error (MAE, the $L1$ counterpart of MSE).

The underlying commonality across all of these functions is that they completely reduce across the spatial dimensions $I$ and $J$. A significant issue with spatial averaging for evaluation is that it becomes unknown precisely where models are and are not performing well. Accordingly, it is impossible to know whether they can be trusted at inference time in a given location. This all despite the fact that erroneously mild temperature predictions increases mortality (Shrader et al., 2023) and previous investigation which shows the existence of temperature prediction disparities in the context of the United States alone (Washington Post, 2024). With SAFE, we uncover spatial disparities in performance across the globe by separating the spatial dimensions into different strata and calculating performance within each. This enables us to understand where models perform best, and facilitates a quantitative fairness evaluation and benchmark across models. Thus, we have two main contributions:

1. The first comprehensive evaluation of fairness in AI weather prediction, demonstrating systemic spatial disparities exist.
2. The introduction of SAFE, a new open-source package that facilitates such evaluations.

> **Takeaway 1**
>
> The state of the art of AI weather prediction relies on spatially-averaged objective functions and evaluation metrics. This masks disparities that exist in where models perform well, despite the fatal consequences of poor predictive power.

## 2 RELATED WORK

### 2.1 REGION STRATIFICATION

WeatherBench 2 (WB2) (Rasp et al., 2024) is an existing benchmark that assesses the spatially-averaged error of models using weather data from ERA5, ECMWF's most modern reanalysis dataset (Hersbach et al., 2020). It provides functionality to get per-region RMSE, but these regions are coarse-grained and exclusively rectangular, making them unusable for the real-world attributes we care about. There are 20 of these regions, 3 that are hemispheric (North, Tropical, and Southern) and 17 geographic. These regions are overlapping and include oceans, but the geographic regions miss considerable sections of populated landmass (including but not limited to significant portions of Central America, Eastern Africa, Brazil, California, and the island of New Guinea). The hemispheric

regions cover the whole globe, with the Tropical region bounded by the $\pm 20°$ latitude lines. GraphCast reports per-region RMSE for T850 and Z500 in Supp. Mat. Fig. S14–S16 on these naively defined regions (Lam et al., 2023). NeuralGCM also only looks at the region attribute, considering only North Atlantic, Northeast Pacific, Northwest Pacific, Other North, South Indian, South Pacific, Other South, Tropics ($\pm 20°$), Extratropics ($[30°, 70°]$) (Kochkov et al., 2024), tropical cyclone tracks in the Atlantic, and the Intertropical Convergence Zone and atmospheric river in the Atlantic. Later, in supplementary material, Kochkov et al. (2024) redefine the tropics as ($\pm 15°$) and the extratropics as ($[25°, 55°]$).

Stable equitable error in probability space (SEEPS) (Rodwell et al., 2010) is a metric that was introduced to assess the quality of precipitation forecasts in particular. In the original paper (Rodwell et al., 2010), the authors perform region-specific analysis of forecasts in South America, Europe, and the extropics. Again, however, the region shapes are defined with crude, rectangular boundaries ([70°W–35°W, 40°S–10°N], [12.5°W–42.5°E, 35°N–75°N], and [above 30°N or below 30°S], respectively).

In contrast, the regions used within SAFE cover all landmass (including islands) across the Earth and are carefully crafted to not include oceanic landcover (i.e., they are not the crude rectangular boundaries used in existing work). This more aptly captures metrics for where fairness in weather forecasts matters most: the places where people live. Our regions are non-overlapping, except at their borders where gridpoint polygons stretch over the border (this being an artifact of finite resolution). Additionally, we include attribute categories hitherto ignored in the literature: territory, income, and landcover.

> **Takeaway 2**
>
> Existing approaches to stratify AIWP model performance are rare and at best utilize crude rectangular boundaries, operating only on the region attribute.

### 2.2 FAIRNESS METRICS

The standard approach to assessing fairness in continuous regression problems is to binarize the outcome based on some threshold for predicted value $\hat{y}$ (Jui & Rivas, 2024; Mehrabi et al., 2021; Barocas et al., 2023), and typically are only used in binary (two strata) settings as noted by Yan et al. (2021). This was the case with the United States' erstwhile four-fifths rule, which enforced fairness in hiring through the difference in binary outcome rates across binary group labels. There is a dearth of work and metrics that measure fairness in the continuous output space of variables as noted by Agarwal et al. (2019). Agarwal et al. (2019) attempts to address this by defining fairness metrics for continuous regression cases where the loss function is 1-Lipschitz under the $\ell_1$ norm. However, with climatic variables, $y$, $\hat{y}$ are not bounded by $[0, 1]$ and often exceed 1 (as is the case with all the variables we explore in this work), and the loss function Equation 1 includes the exponential term $(\hat{y} - y)^2$. This means the fairness metrics of Agarwal et al. (2019) cannot apply. Other works on fairness for continuous regression utilize the difference in average $\hat{y}$ across two strata only (Suárez Ferreira et al., 2025; Fitzsimons et al., 2019; Calders et al., 2013).

In summary, fairness metrics used in both the machine learning literature and high impact legal settings are often simple differences in performance calculated with subtraction (e.g., statistical parity difference, equal opportunity difference). The greatest absolute difference and variance measurements we define in subsubsection 3.2.3 are thus firmly grounded in this existing literature. By considering more than two strata at a time, our metrics push the field further. Furthermore, the modular design of SAFE makes it easily extensible to incorporate future fairness metrics as they are developed.

## 3 SAFE

In this paper we create a framework for performing Stratified Assessments of Forecasts over Earth (SAFE). This tool enables stratification by various geographically-related attributes, allowing the user to see the fine-grained quality of a set of predictions when broken down by the different constituent groups, or strata, of each attribute. We leverage SAFE to benchmark the fairness of existing AIWP models.

> **Takeaway 3**
>
> We introduce SAFE, an open source python library that integrates different data sources and facilitates stratified fairness evaluations of AI weather and climate models.

## 3.1 DATA SOURCES

Within SAFE, we provide the ability to investigate different attributes: territory, global subregion, income, and landcover. The strata within the territory attribute is typically the country which a gridpoint is located within, though there are some sub-national or not universally recognized territories. Territory borders are pulled from the geoBoundaries Global Administrative Database (Runfola et al., 2020). Any gridpoint overlapping with any land will be classified as "land" for the landcover attribute and otherwise as "water". Global subregions follow the United Nation's classifications over territories (United Nations). The income stratum of a gridpoint is one of "high income", "upper-middle income", "lower-middle income", or "low-income" as defined by the World Bank's classification for the gridpoint's encompassing territory (World Bank); the World Bank uses the gross national income (GNI) per capita of the territory, calculated using the Atlas methodology. The polygons associated with each strata are accessed through the MIT-licensed pygeoboundaries_geolab package.[1] This package is a python wrapper for the geoBoundaries Global Administrative Database (Runfola et al., 2020), which itself is made available under a open license CC-BY 4.0.

## 3.2 METHODS

### 3.2.1 STRATIFICATION

Forecasts made over the Earth are associated with specific (longitude, latitude) coordinates, or "gridpoints" on the Earth. Each pair of coordinates is converted into the polygon that is centered on the gridpoint but which covers all the quadrilateral surface area defined by extending its borders to the midpoint with its neighbors in both the longitude and latitude directions. To unify the coordinate system across all integrated data sources, latitude ranges [-90, 90] with index 0 at -90, and longitude [-180, 180] but with index 0 at 0 and a wraparound from 180 to -180 in the middle. This is because polygons and associated attribute metadata sourced from pygeoboundaries_geolab follows this coordinate system, and it is easier to bring the other tabular data into conformance than modify this.

The forecasts for a gridpoint's polygon are associated with all of the strata that have any polygon which intersects it. While this will double count some gridpoints towards different strata, measures are taken so that no single gridpoint counts more than once within a given strata. The double counting that does occur is in line with the philosophy of SAFE, as the alternative is that—without high enough resolution—there will be strata for which no data is recorded, rendering them invisible and left out of fairness assessments. Importantly, this "double counting" is a different phenomenon from the "double penalty" described by Gilleland et al. (2009). In total, there are 231 territory, 23 subregion, 4 income, and 2 landcover strata. Of the 231 territories, 213 have an associated income strata. 76 are classified as high-income, 57 as upper-middle-income, 45 as lower-middle-income, and 34 as low-income. Subregions vary from having 1 territory (Antarctica) to 25 (Caribbean). More details on the strata are in Appendix B.

### 3.2.2 AREA WEIGHTING

In calculating the loss function for training it is common to weight the (squared if $L2$) difference in variable prediction and ground truth by the area of the grid cell the forecast was made at before averaging. This is the case with all six models we assess: GraphCast (Lam et al., 2023) Supp. Mat. Eq. 19, Keisler (Keisler, 2022) section 3.3.3, Pangu-Weather (Bi et al., 2022) Eq. 2, Spherical CNN—which refers to Rasp & Thuerey (2021) for training details, FuXi (Chen et al., 2023) Eq. 2, and NeuralGCM (Kochkov et al., 2024) Supp. Mat. G.4; as well as in other state-of-the-art models including but not limited to GenCast (Price et al., 2023) Supp. Mat. D.4, FGN (Alet et al., 2025) Eq. 5, and FourCastNet 3 (Bonev et al., 2025) Eq. 50. This weight varies with latitude. The reason for latitude weighting is that, when using an equiangular gridding, the gridpoints are closer together near

---

[1] https://github.com/ibhalin/pygeoboundaries

the poles than they are at the equator. This results in a higher density of samples per area at the poles, which left unaccounted for could cause the model to overfit to forecasting polar weather.

Complicating the matter, Earth is an oblate spheroid with an equatorial radius of 6,378,137m and a smaller polar radius of 6,356,752m. However, no existent python library known to the authors takes this into account to get the precise surface area of equiangular grid cells on Earth's surface. The assumption of a spherical Earth yields surface areas near the poles that are still greater than they are in reality, meaning the very problem latitude weighting aims to address persists. The standard solution would be to convert the cells to vector data and get the area of polygons. However, virtually every approach, both training/evaluating (Lam et al., 2023; Keisler, 2022; Bi et al., 2022; Kochkov et al., 2024; Pathak et al., 2022; Bonev et al., 2023; Alet et al., 2025; Bonev et al., 2025) and benchmarking (Rasp et al., 2024; Leeuwenburg et al., 2024), make the simplifying assumption of a perfectly spherical Earth. WB2 takes this approach in computing its metrics as well (Rasp et al., 2024). As part SAFE, we have provided a utility that can get the surface area of grid cells on the Earth while taking into account its oblate geometry. We use the equation for getting the surface area of oblate spheroid caps from Calvimontes (2018) Eq. 49 which builds on the model developed by Whyman & Bormashenko (2009). For testing, the total surface area of the Earth was found with the equation for oblate spheroid surface area from Beyer (1987) p. 131, yielding an approximation of $510,065,604,944,206.145m^2$. A spherical model overestimates the latitude weight (normalized by mean grid cell area) of the polar grid cells (i.e., the most northern or southern grid cells) by 0.7% with $1.5°$ resolution and by 504% with $0.25°$ resolution.

In calculating the RMSE as reported throughout this paper, we use these exact surface areas and get the weights by normalizing the grid cell areas by the mean cell area. This same normalization is used in WB2 (Rasp et al., 2024) and is common in training (Pathak et al., 2022; Bonev et al., 2023).

> **Takeaway 4**
>
> SAFE introduces a new state-of-the-art level of accuracy in latitude weighting, a normalization technique used in virtually all AI weather or climate work.

### 3.2.3 METRICS

**Model performance metrics.** The main metric utilized in SAFE is the latitude-weighted RMSE, which is averaged temporally by initialization time (the timestamp of the climate variables fed into the model) not lead time (the amount of time into the future for which to forecast the state of climate variables at), and averaged spatially within each strata. Unless otherwise specified, reported RMSE refers to this. The anomaly correlation coefficient (ACC) is another evaluation metric that is often used for cross-model comparison. It is the only scale-free metric that is commonly used for this purpose. Like RMSE, ACC is spatially averaged (European Centre for Medium-Range Weather Forecasts, 2025a) and would thus benefit from stratified assessment. The fact that the most popular metrics employ spatial averaging underscores the need for SAFE. We emphasize RMSE in this work under the same rationale as taken by WeatherBench: the similarity between RMSE and the models' training objectives (Rasp et al., 2020). Furthermore, RMSE is the predominant metric reported in the literature. In this work we focus on benchmarking deterministic models. Probabilistic, or ensemble, AIWP models have other metrics that can be used such as the continuous ranked probability score (CRPS), but also are commonly evaluated on the RMSE of the ensemble's average prediction.

We motivate the work of stratified fairness through the spatial disparities that exist in AIWP performance that is visible even on the individual gridpoint level. Figure 1 demonstrates an example of this, showing the unequal performance of GraphCast across the globe at forecasting temperature with $\tau = 72h$. The data visualized in this example can be easily accessed with SAFE through a call to the `safe_earth.metrics.errors.stratified_rmse` function.

**Fairness metrics.** We define two metrics for measuring fairness. Both operate on the level of data for individual variables and individual attributes. To start, the RMSE for a model's performance on the given variable is calculated for each strata within the attribute. To characterize the worst-case disparity of each model, we measure (1) the greatest absolute difference in the per-strata RMSEs. To assess the overall nature of the model, we also measure (2) the variance in per-strata RMSEs. A maximally "fair" model will have a value of 0 for both metrics, as this would mean it

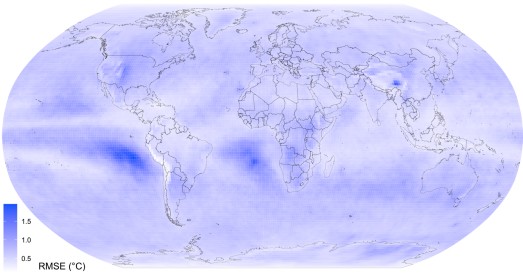

Figure 1: GraphCast displays non-uniform error in temperature prediction. The temporally-averaged gridpoint specific RMSE of temperature predictions at 850hPa (T850) made by GraphCast for every 12 hours in 2020 are shown. Predictions made with 3 day lead time, meaning they predict the temperature 72 hours after the input conditions. Lower RMSE is better. GraphCast inference predictions from WeatherBench 2, ground truth temperature values from ECMWF ERA5. Spatial resolution is 1.5 degrees.

is performing no worse on any strata than any other. These metrics are computed through calls to `safe_earth.metrics.fairness.measure_fairness` within SAFE.

> **Takeaway 5**
>
> We for the first time introduce **fine-grain stratification** in the literature. Current approaches use globally-averaged training objectives and evaluation metrics. To start, SAFE offers stratification on the attributes of territorial affiliations (country), global subregion, income, and landcover (land or water).

## 4 BENCHMARKING AIWP FORECAST FAIRNESS: DEMONSTRATING SAFE

To minimize computational costs, we investigate models with already available predictions. This eliminates the need for model training or inference, reducing the carbon footprint of our research. WB2 provides easily-accessible cloud datasets of ERA5 data and inference runs in the year 2020 for a number of models. Because of the unified access endpoints and resolution, we use the models available through these datasets to begin our investigation. Furthermore, these models are among the most state of the art (by standard metrics such as RMSE and ACC) (Rasp, 2024), so it is in fact preferable to study these than retrain our own, potentially inferior models.

### 4.1 FORECASTS ASSESSED

In this work with utilize WB2's $1.5°$ resolution equiangular predictions on ERA5. We choose the $1.5°$ resolution ($240 \times 121$ in terms of longitude by latitude) because it has the most amount of models with provided forecasts at a single common resolution. The forecasts provided are made on ERA5 data from 2020. WB2 retrieved this subset of ERA5 data from ECMWF via the Copernicus Climate Data Store, which makes its products available through an open license.[2] WB2 itself is available through an Apache License 2.0.

The models evaluated are GraphCast (Lam et al., 2023), Keisler (Keisler, 2022), Pangu-Weather (Bi et al., 2022), Spherical CNN (Esteves et al., 2023), FuXi (Chen et al., 2023), and NeuralGCM (Kochkov et al., 2024); more details on these models are available in Appendix A. All of the assessed models were trained on ERA5 data, making it an appropriate common benchmark, and none of them included 2020 in their training set. The set of lead times $\tau$ that is common to the provided predictions for all models is every 12 hours up to 10 days, so we assess all models at each of these.

---

[2]https://apps.ecmwf.int/datasets/licences/copernicus/

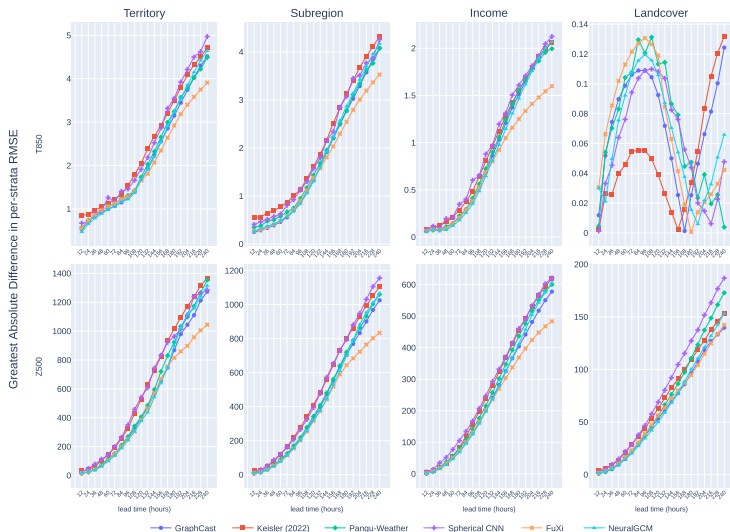

Figure 2: Greatest absolute difference of any two per-strata RMSE for each attribute when predicting T850 and Z500 at different lead times. Lower difference is more fair. Starting at a lead time of about one week, FuXi is the most fair model across all attributes and variables.

## 4.2 VARIABLES

In line with WeatherBench (Rasp et al., 2020; 2024), we choose as our variables $y$ the atmospheric temperature at 850hPa ("T850", unit: K) and geopotential at 500hPa ("Z500", unit: $m^2s^{-2}$) as benchmark variables for comparing cross-model performance in this experiment. Geopotential is the strength of Earth's gravitational field, so predicting the geopotential at a fixed atmospheric pressure level (500hPa) amounts to predicting the vertical synoptic-scale distribution of pressure in Earth's atmosphere. This knowledge is highly useful in meteorological predictions (Lam et al., 2023).

These variables are the most prevalent commonality between different model developers' assessments; that is, they are used by default in reporting model skill for their meteorological importance as outlined above. In their original papers, Pangu-Weather (Bi et al., 2022), Spherical CNN (Esteves et al., 2023), FourCastNet (Pathak et al., 2022), FuXi (Chen et al., 2023), Keisler (Keisler, 2022), and NeuralGCM (Kochkov et al., 2024) are primarily evaluated with T850 and Z500, while GraphCast is an outlier (Lam et al., 2023) reporting mainly on just Z500. We provide results on other variables of interest in Appendix E; those experiments find the same results of systemic bias.

## 4.3 EXPERIMENTAL DESIGN

The main data we collect in our experiments involves taking the area-weighted squared difference between the models prediction $\hat{y}$ and the ERA5 ground truth value $y$ at every individual gridpoint, for every lead time $\tau \in \{12h, 24h, ..., 240h\}$, at every 12 hour interval in 2020.

Then, for each of our four attributes and both variables, we calculate the per-strata RMSE (averaged temporally over the year) at all 20 lead times by taking the RMSE when spatially averaging over only the gridpoints within that strata. This allows us to see which stratum the models are performing best or worst within.

Lastly, for each attribute and variable, we take the greatest absolute difference in per-strata RMSE of any pair of per-strata RMSE with the same attribute and variable. We also take the variance of all the per-strata RMSE to characterize the spread of model performance. This allows us to quantify the fairness of a model's predictions, where the smaller the difference and variance are, the more fair the model. As perfectly fair models are defined as having values of 0 for the metrics we define, the greater these metrics are the more biased a model is considered.

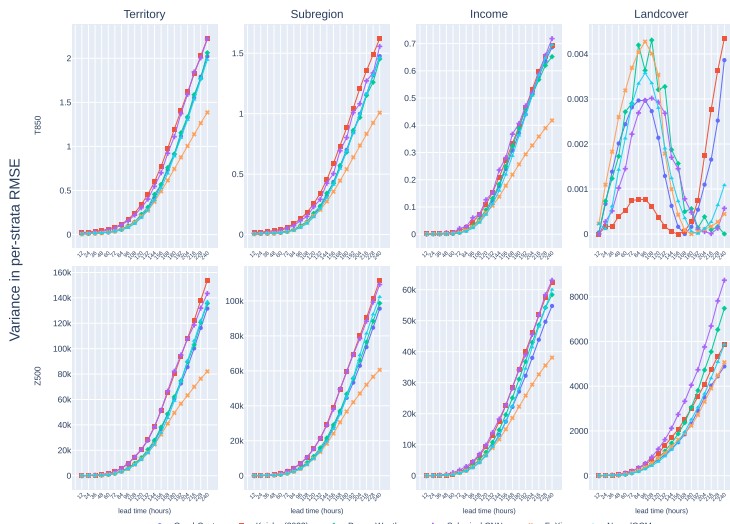

Figure 3: Variance of all the per-strata RMSE for each attribute when predicting T850 and Z500 at different lead times. Lower variance is more fair.

## 4.4 RESULTS

**General fairness.** As seen in Figure 2, the fairness of predictions begin to rapidly decline once the lead time surpasses three days; that is, the greatest absolute difference in RMSE of any two strata rapidly increases. Across all four attributes and all lead times, Spherical CNN and Keisler are generally the least fair. From a lead time of about a week onwards, FuXi is drastically more fair than every other model across all attributes. At early lead times, NeuralGCM appears to perform most fairly.

It is generally seen across all models and variables that the biases in predictions increase exponentially after lead time surpasses a few days. The exception appears to be a dip in bias in the landcover attribute on T850 with roughly eight days of lead time. However, the landcover attribute is a special case where it can be argued that one should care about absolute performance on the land attribute rather than inter-strata fairness. That is, we may want a pro-land bias. This dip corresponds with model's temperature predictions switching from better over land to worse over land than water (Figure 5).

We provide comprehensive benchmarks of the model fairness results in Appendix D. We also calculate the variance in per-strata RMSEs (Figure 3) which displays similar patterns as seen in the greatest absolute difference (Figure 2). The main difference with variance is that it takes a larger lead time for unfairness to start exponentially increasing. Appendix C proves the discovered biases are not driven by outliers. In Appendix E we conduct the same general fairness analysis on additional climatic variables: precipitation (6hr and 24hr cumulative), 2m temperature, and wind speed (U and V components at 10m).

**Income attribute**. To qualitatively characterize the growing unfairness observed in Figure 2, we take a detailed look at the income attribute. Because it only has four strata, it is easy to visualize and meaningful to explore. For lead time $\tau = 12$ hours, Keisler, Pangu-Weather, Spherical CNN, and NeuralGCM perform worst at predicting both variables in low-income territories (Figure 14).[3] However, by $\tau = 48$ hours, every model displays the trend for both variables where prediction skill decreases as income increases; this disparity continues to grow with lead time (Figure 4). This is an interesting result, and it shows that lead time is an important dimension to consider, because the disparity observed at one fixed lead time may not hold at another.

---

[3]The exception being NeuralGCM, where the per-strata RMSE on Z500 for lower-middle-income is 30.60187 versus low-income's 30.58936.

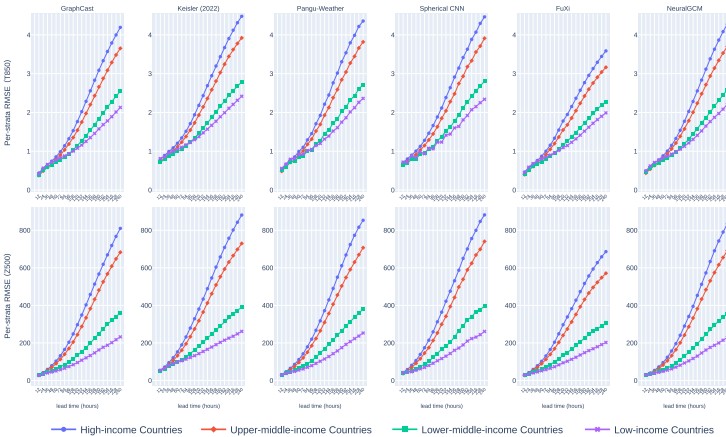

Figure 4: Per-strata RMSE for the income attribute of each model. This captures how well models perform at predicting each climatic variable stratified by the income classification for the associated country. We see that a bias against high income countries grows over time.

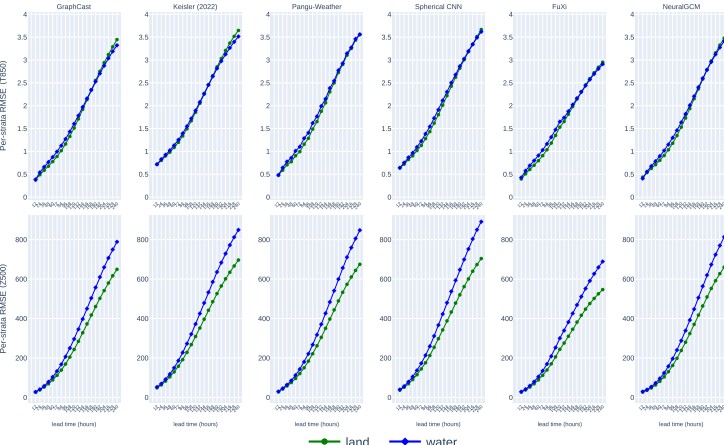

Figure 5: Per-strata RMSE for the landcover attribute of each model. This captures how well models perform at predicting each climatic variable stratified by the prediction being over land or water (oceans, seas, and many large lakes).

**Landcover attribute.** We also take a close look at the landcover attribute. Generally, models perform better over land than water. This can be seen in Figure 5. However, by a lead time of 9 days ($\tau = 216$ hours), all of the models except Pangu-Weather become worse at predicting temperature over land than water. In looking at greatest absolute difference and variance in RMSE, Pangu-Weather did not appear as the most fair with regards to landcover. However, we consider landcover to be a unique attribute. We have a special interest in absolute performance on the land stratum alone as that is where people live (small island nations are dutifully included in "land").[4] In this sense, Pangu-Weather behaves as we may hope in always performing better over land than water, perhaps even more so than if it had equal performance across the strata. This is an exception to the fairness paradigm we laid out before, though it is sensible as all of the other attributes' strata are subsets of the "land" strata. In those cases, we want all the strata to be treated equally. Looking at the landcover attribute as a whole, FuXi is still the most overall fair as at given lead times it has the lowest error for the land stratum.

---

[4]One exception is boats out at sea. For this case, SAFE still provides state of the art advancements as model users can now look at model performance specifically on the oceanic gridpoints they will be traveling across.

> **Takeaway 6**
>
> An example analysis made possible through SAFE **proves the existence of systemic bias in AI weather prediction** by location (at both territory and region resolution), income, and landcover for all climatic variables assessed. The bias increases with lead time.

## 5 FUTURE WORK

An important future direction of work on improving SAFE is incorporating more attributes. Moving beyond binary landcover, work with implicit neural representation (INR) models has shown that it is important to further consider coastlines and islands as their own strata as well (Cai & Balestriero, 2025). Additionally, population density will be added to SAFE as an attribute to better understand the degree to which different AIWP models can be a trusted decision-making tools across different human settlements. This can be done with the LandScan Global dataset (Dobson et al., 2000; Lebakula et al., 2025), and will improve on the already state-of-the-art territory-level precision of this work.

Currently, SAFE operates at inference time. It may prove beneficial to integrate tracking of fairness metrics into the training regimes of models to understand how different training dynamics affect fairness. Further, incorporating spatial stratification into training objectives could ameliorate bias. In general, investigating the underlying causes for why different models are more or less fair and how to remedy this are consequential research questions that are first raised by our work.

## 6 DISCUSSION

Organizations like the NOAA are beginning to incorporate ML systems in their work, citing improvements in models such as ECMWF's very own Artificial Intelligence/Integrated Forecasting System (AIFS) (Konkel, 2024). As AIWP models become increasingly relied upon, the results of this work necessitates more careful attention being paid to the stratified performance and fairness of models. By using SAFE to investigate the territory attribute, one is able to find whether a given AIWP is appropriate to leverage in decision making within that territory. This is an important discovery given the life and death consequences that forecasts can impart. The benchmark provided in this work is a first step in this direction. Moreover, SAFE empowers deployers to select the model which is most performant for their local application given the biases we prove exist. The visibility provided by SAFE into stratified forecast fairness brings this research area to light.

## 7 CONCLUSION

In this work we created SAFE, a python package that allows the user to assess a set of machine learning predictions made over Earth in terms of stratified fairness. Strata are available for four attributes a gridpoint may have: territorial affiliation, global subregion, gross national income per capita, and landcover. This provides developers and decision-makers alike with an important tool to break free from the default approach of spatial averaging. We apply SAFE to a set of state of the art AIWP models, finding that they all display unfair spatial disparities in performance on all four attributes. These disparities generally increase with lead time, particularly after three days. These findings justify our approach of capturing more geographically fine-tuned errors, discouraging the current reliance on spatially-averaged RMSE for characterizing AIWP models. This is an advancement upon the foundation of all AI weather and climate work.

### REPRODUCIBILITY STATEMENT

We have made SAFE open source, including code for reproducing the specific results of section 4, as well as the entire generic framework for promoting similar stratified evaluations on more models and datasets. Code for generating the figures of this work is also included in the repo. We have stylistically altered them in a vector graphics editor, but the data values and representation are the same as those output by the scripts in `demos/` directory. We clearly state the origin of our climate and attribute data in the main text of the paper in subsection 3.1 and subsection 4.1; we also go into

further detail in Appendix B. The code for generating all of the data is part of the SAFE repo in `src/safe_earth/data/`. In subsubsection 3.2.3, we provide code snippets for calculating the metrics we report on.

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

## A  Model details

Original papers, architecture type, and number of parameters (if reported in the original paper) are described in Table 1 for all of the models we asses in our demonstration of SAFE in section 4.

## B  Attribute strata details

Interactive maps showing the geographic locations of each strata for the different attributes is available at <URL REDACTED FOR ANONYMINITY>.

Table 1: Models assessed

| Model | Architecture | Parameters |
|---|---|---|
| GraphCast (Lam et al., 2023) | Graph neural network (GNN) | 36.7 M |
| Keisler (Keisler, 2022) | GNN | 6.7 M |
| Pangu-Weather (Bi et al., 2022) | Earth-specific transformer | 256 M |
| Spherical CNN (Esteves et al., 2023) | Spherical convolutional neural network (CNN) | Not reported |
| FuXi (Chen et al., 2023) | SwinV2 (Liu et al., 2022) transformer blocks in U-net (Ronneberger et al., 2015) arrangement | Not reported |
| NeuralGCM (Kochkov et al., 2024) | Multi-layer perceptrons (MLP) + CNNs + numerical solver | 31.1 M |

## B.1 SUBREGIONS

The 23 strata included in the global subregion attribute are: Antarctica, Australia/New Zealand, Caribbean, Central America, Central Asia, Eastern Africa, Eastern Asia, Eastern Europe, Melanesia, Micronesia, Middle Africa, Northern Africa, Northern America, Northern Europe, Polynesia, South America, South-Eastern Asia, Southern Africa, Southern Asia, Southern Europe, Western Africa, Western Asia, and Western Europe.

## B.2 INCOME

The 18 territory strata without income classifications by the World Bank are: Anguilla; Antarctica; Bonaire, Sint Eustatius, and Saba; Saint Barthelemy; Cook Islands; Falkland Islands; Guadeloupe; French Guiana; Montserrat; Martinique; Mayotte; Niue; Pitcairn Island; Réunion; Saint Helena, Ascension, and Tristan da Cunha; Vatican City; Wallis and Futuna; and Tokelau.

## B.3 LANDCOVER

Due to idiosyncrasies in geoBoundaries, the landcover strata of "land" includes most lakes. Of the 15 largest by surface area, the following are included in "land": Lake Baikal, Great Bear Lake, Great Slave Lake, Lake Winnipeg, Lake Ladoga, and Lake Balkhash. Disambiguation will be improved through the integration of data sources more targeted for landcover. Datasets such as the Generic Mapping Tools (GMT) (Wessel et al., 2019), available in Python with PyGMT (Tian et al., 2025), can provide gridpoint strata of ocean, landmass, lake, land within lake, and lake within land within lake.

## C ACCOUNTING FOR OUTLIERS

For each model we have assessed, the greatest absolute difference and variance in RMSE for each variable decreases as the number of stratum for the attribute decreases. This raises the question of whether the unfairness phenomenon observed results from rare outliers that appear as the geographic area of the smallest stratum decreases. To account for this, we reconduct our general fairness analysis after filtering out the set of outlier per-strata RMSE for every attribute. Because the data is skewed, we do not use Tukey's fences as a determination of outlyingness. Furthermore, as the data is bimodal at high lead times for the territory and subregion attributes, we cannot use the adjusted boxplot (Hubert & Vandervieren, 2008) or adjusted outlyingness (AO) (Hubert & Van der Veeken, 2008) methods either. Thus, we turn to local outlier factor (LOF) (Breunig et al., 2000) as our method of outlier detection. We use the default scikit-learn parameters.

Figure 6 and Figure 7 are the same as Figure 2 and Figure 3, respectively, except the outliers have been filtered out. Because the landcover attribute only has two strata, the notion of an outlier does not make sense and so this attribute has been excluded. To more easily compare the results when both including and excluding outliers, we graph the largest per-strata RMSE as a percent of the smallest per-strata RMSE in Figure 8. While there are slight differences in the greatest absolute difference in RMSE for the territory attribute (as evidenced by the different percentages), the general shape of the curves as a function of lead time holds with minor decreases in amplitude. This shows there are deeper trends in unfairness that are not being driven by outliers alone.

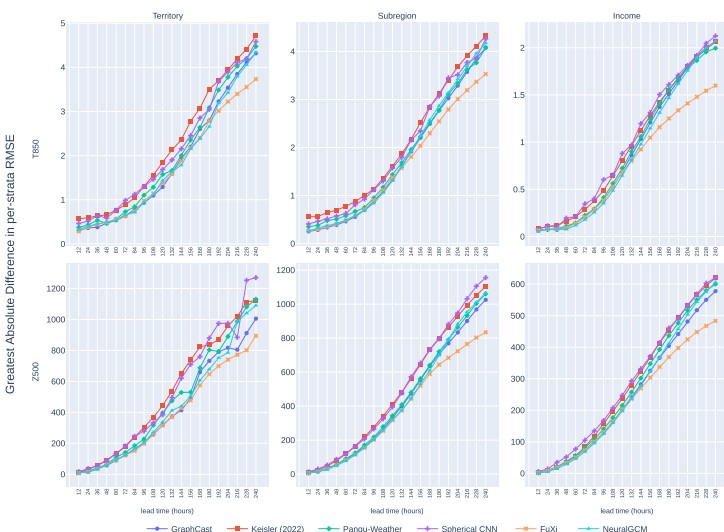

Figure 6: Greatest absolute difference of any two per-strata RMSE for each attribute when predicting T850 and Z500 at different lead times. Lower difference is more fair. Outlier RMSE values have been removed. Starting at a lead time of one week, FuXi is still the most fair model across all attributes and variables.

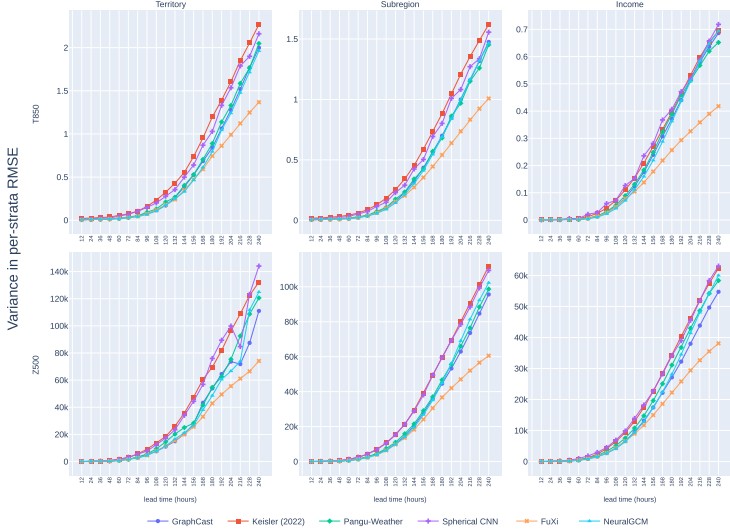

Figure 7: Variance of all the per-strata RMSE for each attribute when predicting T850 and Z500 at different lead times. Lower variance is more fair. Outlier RMSE values have been removed. Starting at a lead time of one week, FuXi is still the most fair model across all attributes and variables.

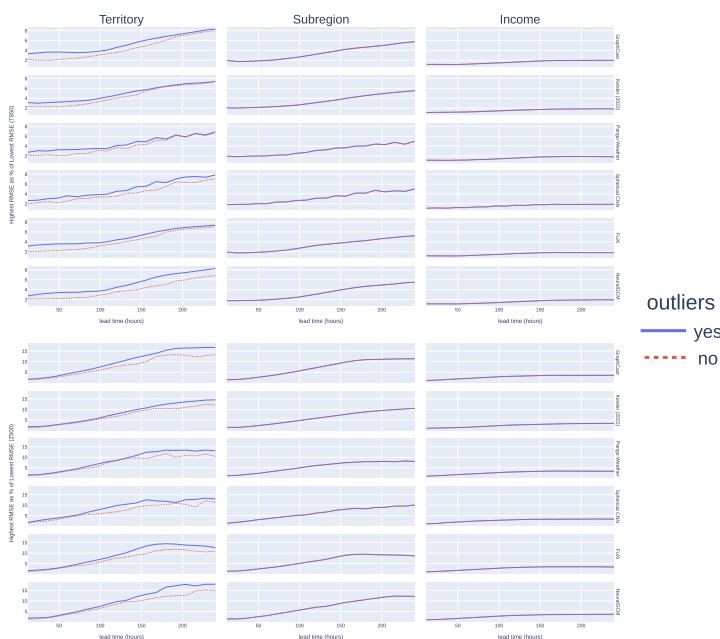

Figure 8: Highest per-strata RMSE as a percent of the lowest per-strata RMSE with and without RMSE outliers filtered out.

## D  Fairness Benchmarks

Table 2: Greatest absolute difference in per-strata RMSE for territory. Lower is more fair; most fair for each variable and lead time is bolded.

| Variable | Lead time (h) | Model | | | | | |
|---|---|---|---|---|---|---|---|
| | | GraphCast | Keisler | Pangu-Weather | Spherical CNN | FuXi | NeuralGCM |
| T850 | 12h | 0.5301 | 0.8523 | 0.5677 | 0.6726 | 0.5548 | **0.4715** |
| T850 | 24h | 0.7129 | 0.8712 | 0.7346 | 0.7011 | 0.7321 | **0.6562** |
| T850 | 36h | 0.8428 | 0.9704 | 0.8578 | 0.9009 | 0.8646 | **0.7861** |
| T850 | 48h | 0.9265 | 1.0467 | 0.9562 | 0.9670 | 0.9731 | **0.8921** |
| T850 | 60h | 0.9991 | 1.1356 | 1.0671 | 1.2620 | 1.0586 | **0.9915** |
| T850 | 72h | 1.0666 | 1.2268 | 1.1138 | 1.1681 | 1.1301 | **1.0552** |
| T850 | 84h | **1.1528** | 1.3355 | 1.2301 | 1.4047 | 1.2289 | 1.1660 |
| T850 | 96h | 1.2629 | 1.5298 | 1.3096 | 1.4621 | 1.2870 | **1.2227** |
| T850 | 108h | 1.3920 | 1.7896 | 1.4194 | 1.6604 | 1.4135 | **1.3685** |
| T850 | 120h | 1.6689 | 2.0537 | 1.7343 | 1.9129 | **1.6558** | 1.6819 |
| T850 | 132h | 1.9286 | 2.3906 | 2.0301 | 2.1865 | **1.8117** | 1.9187 |
| T850 | 144h | 2.2473 | 2.6782 | 2.3223 | 2.5244 | **2.0726** | 2.1994 |
| T850 | 156h | 2.5549 | 2.9147 | 2.6536 | 2.8648 | **2.3466** | 2.5437 |
| T850 | 168h | 2.8560 | 3.2085 | 2.9975 | 3.3171 | **2.6423** | 2.9141 |
| T850 | 180h | 3.1494 | 3.5049 | 3.2451 | 3.5449 | **2.9232** | 3.2373 |
| T850 | 192h | 3.4473 | 3.7977 | 3.5734 | 3.9219 | **3.1848** | 3.5434 |
| T850 | 204h | 3.7448 | 4.0875 | 3.7818 | 4.2116 | **3.4015** | 3.8543 |
| T850 | 216h | 4.0216 | 4.3177 | 4.0281 | 4.4993 | **3.5835** | 4.1497 |
| T850 | 228h | 4.2999 | 4.5044 | 4.2233 | 4.6229 | **3.7494** | 4.4231 |
| T850 | 240h | 4.5130 | 4.7116 | 4.4912 | 4.9728 | **3.9086** | 4.6413 |
| Z500 | 12h | **13.4222** | 31.7980 | 17.1554 | 23.3231 | 15.6101 | 17.7155 |
| Z500 | 24h | 25.4911 | 39.2029 | **22.7217** | 48.5697 | 25.4045 | 27.7922 |
| Z500 | 36h | 44.5530 | 63.3959 | 44.6666 | 78.6476 | 41.8652 | **35.5452** |
| Z500 | 48h | 73.7853 | 101.7007 | 74.2397 | 112.7660 | 67.0475 | **63.8041** |
| Z500 | 60h | 114.5105 | 144.6675 | 112.3888 | 143.2511 | 104.6706 | **98.8060** |
| Z500 | 72h | 149.7483 | 192.5330 | 150.8271 | 199.7432 | 150.0260 | **136.3233** |
| Z500 | 84h | 197.7661 | 254.1886 | 209.0000 | 263.5604 | 199.1568 | **189.1055** |
| Z500 | 96h | 258.1191 | 326.9143 | 268.7369 | 350.5695 | 251.8137 | **243.2503** |
| Z500 | 108h | 327.6491 | 430.2865 | 341.8734 | 457.4359 | **305.2133** | 305.4085 |
| Z500 | 120h | 396.9567 | 528.5107 | 405.2052 | 541.2818 | 386.7127 | **377.4789** |
| Z500 | 132h | 471.5472 | 631.6456 | 485.4492 | 608.7734 | 464.0384 | **437.4839** |
| Z500 | 144h | 566.6680 | 729.6151 | 595.1192 | 743.3179 | 569.2445 | **542.7397** |
| Z500 | 156h | 654.4902 | 822.9236 | 720.2494 | 832.6830 | 669.9283 | **639.2373** |
| Z500 | 168h | 747.2870 | 932.4173 | 830.2883 | 922.7379 | 754.8806 | **747.1750** |
| Z500 | 180h | 868.6595 | 1015.7405 | 923.0644 | 963.3164 | **815.4693** | 898.6278 |
| Z500 | 192h | 979.1869 | 1095.5909 | 1031.9875 | 1014.1516 | **859.9232** | 1023.4053 |
| Z500 | 204h | 1044.4335 | 1171.7235 | 1098.2249 | 1110.0590 | **898.6318** | 1120.0131 |
| Z500 | 216h | 1110.4524 | 1237.9779 | 1173.1740 | 1239.1351 | **958.4204** | 1156.8152 |
| Z500 | 228h | 1212.0978 | 1314.9546 | 1266.0661 | 1261.9217 | **1005.6409** | 1239.9979 |
| Z500 | 240h | 1273.6925 | 1367.5504 | 1354.0935 | 1285.5398 | **1044.5271** | 1315.3407 |

Table 3: Greatest absolute difference in per-strata RMSE for global subregion. Lower is more fair; most fair for each variable and lead time is bolded.

| Variable | Lead time (h) | Model | | | | | |
|---|---|---|---|---|---|---|---|
| | | GraphCast | Keisler | Pangu-Weather | Spherical CNN | FuXi | NeuralGCM |
| T850 | 12h | **0.2525** | 0.5555 | 0.3504 | 0.4085 | 0.2690 | 0.2599 |
| T850 | 24h | **0.2852** | 0.5617 | 0.3779 | 0.4630 | 0.2981 | 0.3243 |
| T850 | 36h | **0.3363** | 0.6454 | 0.4780 | 0.5130 | 0.3512 | 0.3723 |
| T850 | 48h | **0.3880** | 0.6931 | 0.5075 | 0.5734 | 0.4074 | 0.4177 |
| T850 | 60h | **0.4623** | 0.7767 | 0.5796 | 0.6264 | 0.4834 | 0.4862 |
| T850 | 72h | **0.5562** | 0.8714 | 0.6702 | 0.8082 | 0.5863 | 0.5778 |
| T850 | 84h | 0.6994 | 1.0025 | 0.7491 | 0.9256 | 0.7134 | **0.6865** |
| T850 | 96h | 0.8757 | 1.1339 | 0.9501 | 1.1201 | 0.9017 | **0.8381** |
| T850 | 108h | 1.0833 | 1.3553 | 1.1729 | 1.3028 | 1.1138 | **1.0592** |
| T850 | 120h | 1.3296 | 1.6098 | 1.4336 | 1.5700 | 1.3764 | **1.3057** |
| T850 | 132h | 1.6137 | 1.8671 | 1.6829 | 1.7964 | **1.5735** | 1.5992 |
| T850 | 144h | 1.9168 | 2.1653 | 1.9563 | 2.1541 | **1.8085** | 1.9220 |
| T850 | 156h | 2.2034 | 2.5199 | 2.2203 | 2.3437 | **2.0344** | 2.2378 |
| T850 | 168h | 2.4912 | 2.8395 | 2.5085 | 2.8486 | **2.2963** | 2.5705 |
| T850 | 180h | 2.7716 | 3.1276 | 2.7775 | 3.0715 | **2.5446** | 2.8682 |
| T850 | 192h | 3.0267 | 3.4037 | 3.1109 | 3.4501 | **2.7923** | 3.1401 |
| T850 | 204h | 3.2891 | 3.6766 | 3.3345 | 3.5138 | **3.0081** | 3.4233 |
| T850 | 216h | 3.5752 | 3.9070 | 3.6220 | 3.7692 | **3.1939** | 3.6968 |
| T850 | 228h | 3.8532 | 4.1115 | 3.7643 | 3.8652 | **3.3680** | 3.9615 |
| T850 | 240h | 4.0787 | 4.3223 | 4.0785 | 4.2605 | **3.5287** | 4.1710 |
| Z500 | 12h | 10.4583 | 22.0202 | **7.0142** | 13.0860 | 9.6233 | 12.3408 |
| Z500 | 24h | 15.2147 | 23.3950 | 15.3775 | 29.7619 | 14.1225 | **13.2607** |
| Z500 | 36h | 31.6671 | 47.2062 | 35.2885 | 52.6038 | 30.2638 | **27.0468** |
| Z500 | 48h | 55.0884 | 79.2753 | 56.8891 | 87.0013 | 53.2712 | **50.2599** |
| Z500 | 60h | 84.5594 | 118.5651 | 89.9188 | 120.1655 | 81.1725 | **77.4975** |
| Z500 | 72h | 119.6162 | 165.2460 | 124.3231 | 161.8944 | 114.0495 | **113.2731** |
| Z500 | 84h | 159.4383 | 218.3120 | 169.9461 | 206.6974 | **154.0163** | 156.4037 |
| Z500 | 96h | 206.9141 | 276.7293 | 217.2961 | 265.7229 | **201.8156** | 203.6802 |
| Z500 | 108h | 265.2308 | 340.8177 | 278.5598 | 324.6950 | **254.1852** | 257.5620 |
| Z500 | 120h | 330.6048 | 409.0917 | 342.7813 | 394.7158 | **315.9001** | 320.4175 |
| Z500 | 132h | 398.4901 | 481.7423 | 408.0717 | 477.3002 | **374.1447** | 375.0295 |
| Z500 | 144h | 472.8915 | 560.7308 | 480.2798 | 572.4723 | **442.6990** | 445.9015 |
| Z500 | 156h | 553.3959 | 647.4817 | 560.9786 | 652.9538 | **519.2081** | 530.1986 |
| Z500 | 168h | 635.8493 | 731.0427 | 640.5657 | 732.2401 | **589.3926** | 609.7468 |
| Z500 | 180h | 705.7083 | 800.0373 | 720.1483 | 795.3998 | **644.0070** | 698.6875 |
| Z500 | 192h | 769.0193 | 861.8439 | 789.5368 | 880.8382 | **684.0959** | 791.0014 |
| Z500 | 204h | 833.2273 | 928.2371 | 863.7943 | 947.5966 | **722.9894** | 882.7328 |
| Z500 | 216h | 900.5285 | 993.3099 | 932.0408 | 1031.0736 | **765.0671** | 953.4226 |
| Z500 | 228h | 968.6828 | 1051.8553 | 1003.9197 | 1105.4598 | **802.7399** | 1011.1210 |
| Z500 | 240h | 1025.1756 | 1104.4001 | 1060.0999 | 1155.7330 | **833.8870** | 1066.3462 |

Table 4: Greatest absolute difference in per-strata RMSE for income. Lower is more fair; most fair for each variable and lead time is bolded.

| Variable | Lead time (h) | Model | | | | | |
|---|---|---|---|---|---|---|---|
| | | GraphCast | Keisler | Pangu-Weather | Spherical CNN | FuXi | NeuralGCM |
| T850 | 12h | 0.0620 | 0.0825 | 0.0751 | 0.0774 | 0.0642 | **0.0542** |
| T850 | 24h | 0.0754 | 0.1048 | **0.0727** | 0.1140 | 0.0797 | 0.0805 |
| T850 | 36h | **0.0715** | 0.1201 | 0.0813 | 0.0972 | 0.0757 | 0.0758 |
| T850 | 48h | 0.0937 | 0.1607 | 0.1106 | 0.1952 | 0.0928 | **0.0754** |
| T850 | 60h | 0.1423 | 0.2117 | 0.1430 | 0.2091 | 0.1405 | **0.1181** |
| T850 | 72h | 0.2078 | 0.2854 | 0.2241 | 0.3468 | 0.2050 | **0.1791** |
| T850 | 84h | 0.2900 | 0.3773 | 0.2926 | 0.4009 | 0.2881 | **0.2559** |
| T850 | 96h | 0.3875 | 0.4873 | 0.4117 | 0.6024 | 0.3822 | **0.3522** |
| T850 | 108h | 0.5282 | 0.6422 | 0.5619 | 0.6514 | 0.5243 | **0.4859** |
| T850 | 120h | 0.6864 | 0.8056 | 0.7235 | 0.8795 | 0.6860 | **0.6450** |
| T850 | 132h | 0.8621 | 0.9584 | 0.8995 | 0.9707 | **0.8004** | 0.8119 |
| T850 | 144h | 1.0326 | 1.1233 | 1.0585 | 1.1959 | **0.9227** | 0.9777 |
| T850 | 156h | 1.2090 | 1.2809 | 1.2427 | 1.3127 | **1.0478** | 1.1458 |
| T850 | 168h | 1.3726 | 1.4249 | 1.4092 | 1.5043 | **1.1584** | 1.3107 |
| T850 | 180h | 1.5107 | 1.5502 | 1.5566 | 1.6087 | **1.2506** | 1.4676 |
| T850 | 192h | 1.6483 | 1.6787 | 1.6764 | 1.7082 | **1.3365** | 1.6183 |
| T850 | 204h | 1.7905 | 1.7966 | 1.7819 | 1.8130 | **1.4103** | 1.7575 |
| T850 | 216h | 1.9016 | 1.9114 | 1.8654 | 1.9200 | **1.4795** | 1.8932 |
| T850 | 228h | 1.9866 | 2.0008 | 1.9558 | 2.0456 | **1.5437** | 1.9994 |
| T850 | 240h | 2.0647 | 2.0616 | 1.9952 | 2.1247 | **1.5983** | 2.0702 |
| Z500 | 12h | **0.8108** | 3.6642 | 1.6727 | 5.9048 | 1.5137 | 1.6957 |
| Z500 | 24h | 7.2642 | 8.9651 | 8.9447 | 13.8836 | 7.8770 | **5.0367** |
| Z500 | 36h | 18.9145 | 19.0145 | 20.9362 | 34.6394 | 18.7908 | **15.1110** |
| Z500 | 48h | 34.0168 | 34.0692 | 37.4726 | 51.7026 | 33.1263 | **28.8815** |
| Z500 | 60h | 52.3629 | 53.6224 | 56.2572 | 77.0541 | 50.8687 | **46.4533** |
| Z500 | 72h | 74.6146 | 84.2772 | 80.9830 | 104.9152 | 73.3257 | **68.6393** |
| Z500 | 84h | 100.2624 | 118.3149 | 108.8336 | 134.4815 | 100.0320 | **95.4571** |
| Z500 | 96h | 129.4711 | 156.8515 | 140.8503 | 167.4550 | 130.5978 | **124.9093** |
| Z500 | 108h | 163.7621 | 196.7506 | 176.3627 | 207.2879 | 165.2451 | **159.4563** |
| Z500 | 120h | 201.4575 | 237.4061 | 215.1524 | 247.2022 | 202.5977 | **196.7490** |
| Z500 | 132h | 240.7642 | 279.0208 | 257.4120 | 292.4811 | **236.1156** | 236.5664 |
| Z500 | 144h | 282.1859 | 323.2796 | 301.8338 | 331.7819 | **269.7027** | 277.7673 |
| Z500 | 156h | 325.4654 | 367.9642 | 347.4286 | 370.2875 | **304.1304** | 323.0253 |
| Z500 | 168h | 366.2331 | 413.1306 | 392.5974 | 414.4043 | **337.6676** | 365.8929 |
| Z500 | 180h | 404.1501 | 454.5309 | 436.6129 | 460.6266 | **368.8637** | 410.7149 |
| Z500 | 192h | 441.7336 | 493.7207 | 476.3061 | 490.5075 | **397.4898** | 457.3617 |
| Z500 | 204h | 481.4574 | 531.8207 | 516.7449 | 532.7525 | **424.5340** | 503.4393 |
| Z500 | 216h | 517.2955 | 566.0298 | 550.1209 | 568.5136 | **448.0975** | 543.1820 |
| Z500 | 228h | 550.2756 | 594.4706 | 579.6324 | 602.9630 | **467.4228** | 576.0455 |
| Z500 | 240h | 577.7541 | 619.7738 | 600.2285 | 620.6610 | **483.7225** | 606.3814 |

Table 5: Greatest absolute difference in per-strata RMSE for landcover. Lower is more fair; most fair for each variable and lead time is bolded.

| | | Model | | | | | |
|---|---|---|---|---|---|---|---|
| Variable | Lead time (h) | GraphCast | Keisler | Pangu-Weather | Spherical CNN | FuXi | NeuralGCM |
| T850 | 12h | 0.0119 | 0.0022 | 0.0047 | **0.0014** | 0.0305 | 0.0301 |
| T850 | 24h | 0.0518 | 0.0262 | 0.0542 | 0.0331 | 0.0662 | **0.0211** |
| T850 | 36h | 0.0744 | **0.0258** | 0.0703 | 0.0454 | 0.0855 | 0.0497 |
| T850 | 48h | 0.0897 | **0.0398** | 0.0831 | 0.0640 | 0.1020 | 0.0756 |
| T850 | 60h | 0.0988 | **0.0456** | 0.1042 | 0.0762 | 0.1130 | 0.0922 |
| T850 | 72h | 0.1061 | **0.0546** | 0.1077 | 0.0942 | 0.1215 | 0.1076 |
| T850 | 84h | 0.1089 | **0.0554** | 0.1296 | 0.1037 | 0.1271 | 0.1156 |
| T850 | 96h | 0.1090 | **0.0555** | 0.1206 | 0.1090 | 0.1308 | 0.1196 |
| T850 | 108h | 0.1045 | **0.0497** | 0.1313 | 0.1099 | 0.1266 | 0.1158 |
| T850 | 120h | 0.0925 | **0.0389** | 0.1132 | 0.1083 | 0.1189 | 0.1059 |
| T850 | 132h | 0.0718 | **0.0263** | 0.1144 | 0.1036 | 0.0845 | 0.0873 |
| T850 | 144h | 0.0500 | **0.0140** | 0.0865 | 0.0825 | 0.0631 | 0.0705 |
| T850 | 156h | 0.0254 | **0.0024** | 0.0791 | 0.0762 | 0.0415 | 0.0541 |
| T850 | 168h | **0.0014** | 0.0160 | 0.0446 | 0.0560 | 0.0190 | 0.0376 |
| T850 | 180h | 0.0253 | 0.0335 | 0.0475 | 0.0439 | **0.0007** | 0.0155 |
| T850 | 192h | 0.0465 | 0.0549 | 0.0235 | 0.0201 | 0.0139 | **0.0059** |
| T850 | 204h | 0.0663 | 0.0833 | 0.0392 | **0.0147** | 0.0206 | 0.0219 |
| T850 | 216h | 0.0814 | 0.1052 | 0.0195 | **0.0062** | 0.0259 | 0.0327 |
| T850 | 228h | 0.1004 | 0.1206 | 0.0256 | **0.0227** | 0.0328 | 0.0507 |
| T850 | 240h | 0.1243 | 0.1318 | **0.0039** | 0.0478 | 0.0423 | 0.0659 |
| Z500 | 12h | 1.1498 | 4.0162 | **0.9773** | 2.4792 | 1.6285 | 1.8373 |
| Z500 | 24h | **2.5139** | 5.8507 | 2.7293 | 5.1420 | 3.3727 | 2.9665 |
| Z500 | 36h | **5.1433** | 8.9834 | 5.8332 | 9.6810 | 6.1696 | 5.7374 |
| Z500 | 48h | **9.4250** | 13.8543 | 10.1021 | 14.7946 | 10.4900 | 9.5702 |
| Z500 | 60h | 14.8999 | 20.7261 | 16.1954 | 21.7804 | 16.0543 | **14.4554** |
| Z500 | 72h | 21.4036 | 28.3228 | 22.4379 | 28.5432 | 22.7172 | **20.1387** |
| Z500 | 84h | 28.8944 | 36.0126 | 29.9796 | 37.6959 | 30.2328 | **27.1628** |
| Z500 | 96h | 37.2341 | 44.2808 | 37.3130 | 46.6026 | 38.5729 | **34.4391** |
| Z500 | 108h | 45.4507 | 53.2869 | 45.8862 | 57.6306 | 47.2646 | **41.9945** |
| Z500 | 120h | 53.2044 | 63.1831 | 55.7587 | 68.9327 | 56.2924 | **49.9061** |
| Z500 | 132h | 61.3276 | 73.3235 | 66.3274 | 80.2869 | 63.5645 | **58.8711** |
| Z500 | 144h | 69.6294 | 82.5196 | 76.2810 | 92.0807 | 72.1485 | **68.2075** |
| Z500 | 156h | 77.3385 | 91.0754 | 86.4044 | 104.6094 | 80.2376 | **77.3269** |
| Z500 | 168h | **85.9806** | 100.1707 | 97.3472 | 115.3360 | 87.0638 | 88.4507 |
| Z500 | 180h | 96.4382 | 109.5743 | 110.5671 | 127.2699 | **94.6042** | 100.2435 |
| Z500 | 192h | 107.7786 | 118.8574 | 123.4246 | 137.6266 | **104.2935** | 110.5522 |
| Z500 | 204h | 118.3307 | 127.7590 | 137.4269 | 151.7028 | **114.9430** | 121.0782 |
| Z500 | 216h | 126.9011 | 137.7304 | 148.8532 | 163.5775 | **124.9670** | 131.9841 |
| Z500 | 228h | **133.3081** | 146.0886 | 161.5919 | 176.7622 | 133.9938 | 142.6224 |
| Z500 | 240h | **139.7696** | 152.9192 | 172.9741 | 186.9395 | 142.3599 | 153.2736 |

Table 6: Variance of per-strata RMSE for territory. Lower is more fair; most fair for each variable and lead time is bolded.

| Variable | Lead time (h) | Model | | | | | |
|---|---|---|---|---|---|---|---|
| | | GraphCast | Keisler | Pangu-Weather | Spherical CNN | FuXi | NeuralGCM |
| T850 | 12h | 0.0059 | 0.0239 | 0.0092 | 0.0165 | **0.0058** | 0.0076 |
| T850 | 24h | **0.0096** | 0.0279 | 0.0121 | 0.0168 | 0.0097 | 0.0107 |
| T850 | 36h | **0.0135** | 0.0382 | 0.0186 | 0.0307 | 0.0138 | 0.0142 |
| T850 | 48h | **0.0178** | 0.0466 | 0.0221 | 0.0323 | 0.0184 | 0.0183 |
| T850 | 60h | 0.0251 | 0.0620 | 0.0299 | 0.0547 | 0.0255 | **0.0251** |
| T850 | 72h | 0.0373 | 0.0835 | 0.0443 | 0.0716 | 0.0372 | **0.0356** |
| T850 | 84h | 0.0573 | 0.1176 | 0.0607 | 0.1101 | 0.0561 | **0.0524** |
| T850 | 96h | 0.0893 | 0.1698 | 0.0994 | 0.1556 | 0.0876 | **0.0795** |
| T850 | 108h | 0.1375 | 0.2443 | 0.1423 | 0.2218 | 0.1364 | **0.1235** |
| T850 | 120h | 0.2059 | 0.3395 | 0.2291 | 0.3035 | 0.2074 | **0.1919** |
| T850 | 132h | 0.2959 | 0.4561 | 0.3103 | 0.4021 | **0.2714** | 0.2779 |
| T850 | 144h | 0.4186 | 0.5987 | 0.4553 | 0.5464 | **0.3716** | 0.3909 |
| T850 | 156h | 0.5656 | 0.7738 | 0.5670 | 0.7006 | **0.4904** | 0.5290 |
| T850 | 168h | 0.7351 | 0.9761 | 0.7595 | 0.9217 | **0.6140** | 0.6940 |
| T850 | 180h | 0.9111 | 1.1959 | 0.9136 | 1.1105 | **0.7442** | 0.8872 |
| T850 | 192h | 1.1169 | 1.4065 | 1.1541 | 1.3722 | **0.8763** | 1.0970 |
| T850 | 204h | 1.3242 | 1.6228 | 1.3345 | 1.5895 | **1.0046** | 1.3022 |
| T850 | 216h | 1.5543 | 1.8264 | 1.5862 | 1.8438 | **1.1379** | 1.5319 |
| T850 | 228h | 1.7904 | 2.0273 | 1.7694 | 2.0041 | **1.2647** | 1.7675 |
| T850 | 240h | 2.0216 | 2.2219 | 2.0619 | 2.2228 | **1.3857** | 1.9841 |
| Z500 | 12h | 6.1246 | 19.7576 | 6.5390 | 26.6958 | **5.4692** | 8.3412 |
| Z500 | 24h | 32.5049 | 83.0658 | 35.6558 | 119.9938 | 27.7501 | **19.6303** |
| Z500 | 36h | 143.1572 | 279.3724 | 140.8770 | 373.2934 | 120.7857 | **93.2486** |
| Z500 | 48h | 422.6683 | 763.1224 | 392.4690 | 853.2862 | 369.6252 | **299.1150** |
| Z500 | 60h | 987.4937 | 1701.4127 | 949.6804 | 1737.7194 | 876.4381 | **762.7267** |
| Z500 | 72h | 1922.5214 | 3308.0522 | 1841.8360 | 3359.2755 | 1730.4532 | **1597.9239** |
| Z500 | 84h | 3460.4605 | 5910.8440 | 3581.2083 | 5685.9188 | 3114.8778 | **3035.5692** |
| Z500 | 96h | 5878.2943 | 9701.7932 | 5924.0644 | 9238.9938 | 5365.4816 | **5199.0114** |
| Z500 | 108h | 9207.0815 | 14545.4886 | 9712.0473 | 14644.0367 | 8691.7610 | **8420.0934** |
| Z500 | 120h | 13458.0770 | 20514.0618 | 14100.7981 | 20498.3937 | 13073.7371 | **12613.6073** |
| Z500 | 132h | 19099.0530 | 28397.0361 | 20510.6848 | 27854.1528 | 18151.6719 | **17898.7947** |
| Z500 | 144h | 26886.4435 | 38804.6110 | 27905.5223 | 38417.8133 | **24577.2698** | 24711.4568 |
| Z500 | 156h | 36642.9245 | 51445.5371 | 38171.4196 | 51472.3563 | **32413.7343** | 33955.2442 |
| Z500 | 168h | 48409.7383 | 65480.8310 | 48605.6695 | 66563.1440 | **41054.8529** | 45401.1447 |
| Z500 | 180h | 60518.6516 | 80059.5577 | 62042.4154 | 82184.3559 | **49440.1657** | 58908.7073 |
| Z500 | 192h | 72613.8939 | 93940.8663 | 74645.1235 | 94987.2438 | **56671.8077** | 74045.3921 |
| Z500 | 204h | 85533.4150 | 108180.5635 | 89582.7398 | 107801.6592 | **63250.6773** | 90607.4881 |
| Z500 | 216h | 100214.9498 | 122220.9953 | 103077.6203 | 118576.9672 | **69991.3504** | 106317.4791 |
| Z500 | 228h | 116333.6458 | 137609.8284 | 120685.8377 | 131908.1464 | **76345.2022** | 122233.5907 |
| Z500 | 240h | 131597.0478 | 153879.3978 | 135534.6276 | 143461.2929 | **82097.1713** | 137038.3218 |

Table 7: Variance of per-strata RMSE for global subregion. Lower is more fair; most fair for each variable and lead time is bolded. Smallest value determined before rounding to fourth decimal digit for display.

| Variable | Lead time (h) | Model | | | | | |
|---|---|---|---|---|---|---|---|
| | | GraphCast | Keisler | Pangu-Weather | Spherical CNN | FuXi | NeuralGCM |
| T850 | 12h | **0.0040** | 0.0191 | 0.0074 | 0.0108 | 0.0040 | 0.0054 |
| T850 | 24h | **0.0058** | 0.0195 | 0.0083 | 0.0128 | 0.0058 | 0.0073 |
| T850 | 36h | 0.0076 | 0.0266 | 0.0123 | 0.0170 | **0.0075** | 0.0093 |
| T850 | 48h | 0.0099 | 0.0321 | 0.0142 | 0.0237 | **0.0096** | 0.0112 |
| T850 | 60h | 0.0147 | 0.0432 | 0.0184 | 0.0326 | **0.0140** | 0.0150 |
| T850 | 72h | 0.0240 | 0.0603 | 0.0297 | 0.0528 | 0.0226 | **0.0225** |
| T850 | 84h | 0.0400 | 0.0885 | 0.0417 | 0.0735 | 0.0377 | **0.0355** |
| T850 | 96h | 0.0654 | 0.1291 | 0.0742 | 0.1152 | 0.0627 | **0.0577** |
| T850 | 108h | 0.1031 | 0.1828 | 0.1060 | 0.1500 | 0.0997 | **0.0917** |
| T850 | 120h | 0.1575 | 0.2533 | 0.1753 | 0.2296 | 0.1554 | **0.1441** |
| T850 | 132h | 0.2307 | 0.3424 | 0.2346 | 0.2908 | **0.2041** | 0.2146 |
| T850 | 144h | 0.3239 | 0.4540 | 0.3420 | 0.4245 | **0.2741** | 0.3044 |
| T850 | 156h | 0.4354 | 0.5892 | 0.4239 | 0.5038 | **0.3547** | 0.4125 |
| T850 | 168h | 0.5633 | 0.7316 | 0.5709 | 0.6931 | **0.4451** | 0.5456 |
| T850 | 180h | 0.7000 | 0.8827 | 0.6815 | 0.8021 | **0.5407** | 0.6897 |
| T850 | 192h | 0.8463 | 1.0458 | 0.8625 | 1.0085 | **0.6392** | 0.8390 |
| T850 | 204h | 0.9986 | 1.2096 | 0.9686 | 1.0821 | **0.7371** | 0.9964 |
| T850 | 216h | 1.1588 | 1.3546 | 1.1507 | 1.2710 | **0.8334** | 1.1658 |
| T850 | 228h | 1.3176 | 1.4905 | 1.2595 | 1.3367 | **0.9237** | 1.3212 |
| T850 | 240h | 1.4755 | 1.6207 | 1.4524 | 1.5565 | **1.0082** | 1.4636 |
| Z500 | 12h | 4.3923 | 21.0451 | **2.8972** | 13.7473 | 4.1863 | 7.5274 |
| Z500 | 24h | 20.3463 | 51.3661 | 23.1773 | 69.7545 | 16.2691 | **11.0202** |
| Z500 | 36h | 100.9786 | 180.1252 | 103.6128 | 257.4925 | 81.9689 | **58.8206** |
| Z500 | 48h | 306.4674 | 508.7010 | 311.7182 | 654.5374 | 260.7049 | **211.2787** |
| Z500 | 60h | 716.7357 | 1184.5767 | 745.8135 | 1328.9283 | 627.4897 | **550.1721** |
| Z500 | 72h | 1409.4286 | 2363.1666 | 1476.1021 | 2530.5781 | 1266.1543 | **1185.5958** |
| Z500 | 84h | 2530.5646 | 4270.6684 | 2772.6681 | 4229.7121 | 2300.8085 | **2260.5735** |
| Z500 | 96h | 4226.6600 | 7043.0612 | 4561.6586 | 6801.1818 | 3926.2459 | **3892.6980** |
| Z500 | 108h | 6720.7189 | 10643.8530 | 7468.2495 | 10438.1276 | 6378.4510 | **6359.9607** |
| Z500 | 120h | 10141.6595 | 15290.3272 | 10987.4233 | 15390.8119 | **9782.2574** | 9793.6826 |
| Z500 | 132h | 14620.2634 | 21372.4804 | 15893.7948 | 21336.5717 | **13540.9848** | 14039.6015 |
| Z500 | 144h | 20488.9905 | 29376.4397 | 21538.2609 | 28923.9358 | **18291.9364** | 19539.8322 |
| Z500 | 156h | 27792.6869 | 39121.5091 | 29073.4592 | 38107.4375 | **24161.9448** | 26805.0344 |
| Z500 | 168h | 36070.0516 | 49415.4562 | 37029.6547 | 49476.3792 | **30594.7232** | 35135.2208 |
| Z500 | 180h | 44563.2319 | 59680.5409 | 46715.6210 | 59530.9165 | **36729.6627** | 44896.8645 |
| Z500 | 192h | 53264.2736 | 69510.8777 | 55549.5099 | 69291.2225 | **42029.8539** | 56232.0981 |
| Z500 | 204h | 62941.2933 | 80014.6105 | 65986.8602 | 78463.7736 | **47054.4341** | 69073.4405 |
| Z500 | 216h | 73576.4355 | 90367.1381 | 76461.9376 | 88426.2503 | **52021.9554** | 81403.3079 |
| Z500 | 228h | 84684.8208 | 101170.3770 | 88451.4026 | 99330.6919 | **56592.5334** | 92347.6782 |
| Z500 | 240h | 95626.2297 | 111864.2591 | 98694.2737 | 109405.8148 | **60587.5795** | 102322.9160 |

Table 8: Variance of per-strata RMSE for income. Lower is more fair; most fair for each variable and lead time is bolded. Smallest value determined before rounding to fourth decimal digit for display.

| Variable | Lead time (h) | Model | | | | | |
|---|---|---|---|---|---|---|---|
| | | GraphCast | Keisler | Pangu-Weather | Spherical CNN | FuXi | NeuralGCM |
| T850 | 12h | 0.0006 | 0.0013 | 0.0009 | 0.0011 | 0.0007 | **0.0006** |
| T850 | 24h | 0.0010 | 0.0016 | 0.0011 | 0.0017 | 0.0011 | **0.0009** |
| T850 | 36h | 0.0011 | 0.0019 | 0.0013 | 0.0016 | 0.0012 | **0.0010** |
| T850 | 48h | 0.0013 | 0.0033 | 0.0016 | 0.0056 | 0.0014 | **0.0010** |
| T850 | 60h | 0.0026 | 0.0061 | 0.0026 | 0.0057 | 0.0025 | **0.0018** |
| T850 | 72h | 0.0062 | 0.0125 | 0.0073 | 0.0212 | 0.0059 | **0.0044** |
| T850 | 84h | 0.0139 | 0.0243 | 0.0145 | 0.0273 | 0.0135 | **0.0106** |
| T850 | 96h | 0.0276 | 0.0438 | 0.0312 | 0.0603 | 0.0265 | **0.0227** |
| T850 | 108h | 0.0488 | 0.0724 | 0.0543 | 0.0721 | 0.0478 | **0.0426** |
| T850 | 120h | 0.0798 | 0.1103 | 0.0893 | 0.1264 | 0.0785 | **0.0716** |
| T850 | 132h | 0.1238 | 0.1542 | 0.1313 | 0.1532 | **0.1045** | 0.1113 |
| T850 | 144h | 0.1762 | 0.2094 | 0.1835 | 0.2357 | **0.1380** | 0.1600 |
| T850 | 156h | 0.2393 | 0.2702 | 0.2485 | 0.2810 | **0.1782** | 0.2193 |
| T850 | 168h | 0.3072 | 0.3337 | 0.3235 | 0.3675 | **0.2187** | 0.2877 |
| T850 | 180h | 0.3717 | 0.3958 | 0.3883 | 0.4064 | **0.2567** | 0.3626 |
| T850 | 192h | 0.4403 | 0.4646 | 0.4587 | 0.4712 | **0.2931** | 0.4383 |
| T850 | 204h | 0.5147 | 0.5308 | 0.5126 | 0.5191 | **0.3260** | 0.5095 |
| T850 | 216h | 0.5801 | 0.5981 | 0.5677 | 0.5882 | **0.3587** | 0.5835 |
| T850 | 228h | 0.6361 | 0.6526 | 0.6199 | 0.6570 | **0.3897** | 0.6473 |
| T850 | 240h | 0.6864 | 0.6947 | 0.6519 | 0.7183 | **0.4179** | 0.6952 |
| Z500 | 12h | **0.1028** | 2.0068 | 0.4295 | 4.7712 | 0.3603 | 0.6834 |
| Z500 | 24h | 8.1396 | 11.1313 | 11.5812 | 32.1529 | 9.3156 | **3.6432** |
| Z500 | 36h | 58.2438 | 50.2198 | 68.4765 | 189.2996 | 56.8211 | **36.4330** |
| Z500 | 48h | 189.4209 | 184.8858 | 223.4966 | 442.7660 | 179.4246 | **136.8048** |
| Z500 | 60h | 451.0046 | 520.4977 | 519.4968 | 974.0071 | 428.2830 | **358.7665** |
| Z500 | 72h | 911.0966 | 1229.6501 | 1080.3229 | 1821.1763 | 883.6421 | **785.7836** |
| Z500 | 84h | 1646.9627 | 2382.9868 | 1968.3536 | 2952.4151 | 1629.0901 | **1513.8882** |
| Z500 | 96h | 2737.5665 | 4144.1104 | 3276.2421 | 4528.4890 | 2747.9434 | **2587.6919** |
| Z500 | 108h | 4348.0463 | 6467.1286 | 5135.8483 | 6914.8034 | 4382.9351 | **4201.6065** |
| Z500 | 120h | 6597.2991 | 9373.6499 | 7560.5792 | 9862.9344 | 6573.8777 | **6412.9097** |
| Z500 | 132h | 9533.7548 | 12964.9450 | 10791.6058 | 13911.4043 | **8959.9675** | 9339.7532 |
| Z500 | 144h | 13148.0557 | 17427.2140 | 14753.4857 | 18254.7028 | **11740.4081** | 12892.9825 |
| Z500 | 156h | 17445.2287 | 22671.1704 | 19650.6572 | 22708.4878 | **14994.6352** | 17418.3039 |
| Z500 | 168h | 22182.4991 | 28490.4680 | 25084.7184 | 28230.3578 | **18601.2094** | 22431.9354 |
| Z500 | 180h | 27146.0603 | 34351.0365 | 31108.3918 | 34170.3147 | **22273.9916** | 28154.0990 |
| Z500 | 192h | 32239.4210 | 40199.3865 | 36779.7222 | 38994.5403 | **25839.6953** | 34533.8555 |
| Z500 | 204h | 37995.0056 | 46201.5532 | 42988.3711 | 45546.9357 | **29384.0512** | 41433.0906 |
| Z500 | 216h | 43861.1521 | 52094.4753 | 48642.4069 | 51911.1326 | **32665.9497** | 48114.7002 |
| Z500 | 228h | 49661.7138 | 57430.4755 | 54233.6775 | 58344.7264 | **35546.9475** | 54187.6957 |
| Z500 | 240h | 54749.8350 | 62421.6302 | 58373.7147 | 63053.7066 | **38117.6736** | 60038.5224 |

Table 9: Variance of per-strata RMSE for landcover. Lower is more fair; most fair for each variable and lead time is bolded. Smallest value determined before rounding to fourth decimal digit for display.

| Variable | Lead time (h) | Model | | | | | |
| | | GraphCast | Keisler | Pangu-Weather | Spherical CNN | FuXi | NeuralGCM |
|---|---|---|---|---|---|---|---|
| T850 | 12h | 0.0000 | 0.0000 | 0.0000 | **0.0000** | 0.0002 | 0.0002 |
| T850 | 24h | 0.0007 | 0.0002 | 0.0007 | 0.0003 | 0.0011 | **0.0001** |
| T850 | 36h | 0.0014 | **0.0002** | 0.0012 | 0.0005 | 0.0018 | 0.0006 |
| T850 | 48h | 0.0020 | **0.0004** | 0.0017 | 0.0010 | 0.0026 | 0.0014 |
| T850 | 60h | 0.0024 | **0.0005** | 0.0027 | 0.0015 | 0.0032 | 0.0021 |
| T850 | 72h | 0.0028 | **0.0007** | 0.0029 | 0.0022 | 0.0037 | 0.0029 |
| T850 | 84h | 0.0030 | **0.0008** | 0.0042 | 0.0027 | 0.0040 | 0.0033 |
| T850 | 96h | 0.0030 | **0.0008** | 0.0036 | 0.0030 | 0.0043 | 0.0036 |
| T850 | 108h | 0.0027 | **0.0006** | 0.0043 | 0.0030 | 0.0040 | 0.0034 |
| T850 | 120h | 0.0021 | **0.0004** | 0.0032 | 0.0029 | 0.0035 | 0.0028 |
| T850 | 132h | 0.0013 | **0.0002** | 0.0033 | 0.0027 | 0.0018 | 0.0019 |
| T850 | 144h | 0.0006 | **0.0000** | 0.0019 | 0.0017 | 0.0010 | 0.0012 |
| T850 | 156h | 0.0002 | **0.0000** | 0.0016 | 0.0014 | 0.0004 | 0.0007 |
| T850 | 168h | **0.0000** | 0.0001 | 0.0005 | 0.0008 | 0.0001 | 0.0004 |
| T850 | 180h | 0.0002 | 0.0003 | 0.0006 | 0.0005 | **0.0000** | 0.0001 |
| T850 | 192h | 0.0005 | 0.0008 | 0.0001 | 0.0001 | 0.0000 | **0.0000** |
| T850 | 204h | 0.0011 | 0.0017 | 0.0004 | **0.0001** | 0.0001 | 0.0001 |
| T850 | 216h | 0.0017 | 0.0028 | 0.0001 | **0.0000** | 0.0002 | 0.0003 |
| T850 | 228h | 0.0025 | 0.0036 | 0.0002 | **0.0001** | 0.0003 | 0.0006 |
| T850 | 240h | 0.0039 | 0.0043 | **0.0000** | 0.0006 | 0.0004 | 0.0011 |
| Z500 | 12h | 0.3305 | 4.0325 | **0.2388** | 1.5367 | 0.6630 | 0.8439 |
| Z500 | 24h | **1.5799** | 8.5578 | 1.8622 | 6.6101 | 2.8439 | 2.2000 |
| Z500 | 36h | **6.6135** | 20.1753 | 8.5065 | 23.4304 | 9.5159 | 8.2295 |
| Z500 | 48h | **22.2078** | 47.9855 | 25.5133 | 54.7201 | 27.5102 | 22.8971 |
| Z500 | 60h | 55.5015 | 107.3930 | 65.5437 | 118.5960 | 64.4349 | **52.2397** |
| Z500 | 72h | 114.5290 | 200.5459 | 125.8643 | 203.6786 | 129.0179 | **101.3914** |
| Z500 | 84h | 208.7222 | 324.2265 | 224.6944 | 355.2449 | 228.5063 | **184.4539** |
| Z500 | 96h | 346.5953 | 490.1982 | 348.0649 | 542.9503 | 371.9671 | **296.5129** |
| Z500 | 108h | 516.4417 | 709.8745 | 526.3853 | 830.3223 | 558.4856 | **440.8835** |
| Z500 | 120h | 707.6771 | 998.0250 | 777.2581 | 1187.9308 | 792.2084 | **622.6550** |
| Z500 | 132h | 940.2671 | 1344.0848 | 1099.8318 | 1611.4985 | 1010.1114 | **866.4512** |
| Z500 | 144h | 1212.0642 | 1702.3716 | 1454.6969 | 2119.7129 | 1301.3511 | **1163.0654** |
| Z500 | 156h | 1495.3124 | 2073.6823 | 1866.4296 | 2735.7801 | 1609.5192 | **1494.8609** |
| Z500 | 168h | **1848.1638** | 2508.5428 | 2369.1183 | 3325.6011 | 1895.0281 | 1955.8816 |
| Z500 | 180h | 2325.0811 | 3001.6319 | 3056.2699 | 4049.4052 | **2237.4910** | 2512.1880 |
| Z500 | 192h | 2904.0572 | 3531.7694 | 3808.4100 | 4735.2684 | **2719.2835** | 3055.4490 |
| Z500 | 204h | 3500.5399 | 4080.5920 | 4721.5363 | 5753.4311 | **3302.9746** | 3664.9805 |
| Z500 | 216h | 4025.9693 | 4742.4189 | 5539.3190 | 6689.3969 | **3904.1884** | 4354.9488 |
| Z500 | 228h | **4442.7599** | 5335.4729 | 6527.9828 | 7811.2172 | 4488.5863 | 5085.2891 |
| Z500 | 240h | **4883.8882** | 5846.0681 | 7480.0098 | 8736.5929 | 5066.5870 | 5873.1976 |

# E    GENERAL FAIRNESS EXPERIMENTS ON ADDITIONAL VARIABLES

We repeat the general fairness experiments from section 4 for precipitation (6hr and 24hr cumulative), 2m temperature, and wind speed (U and V components at 10m). Forecast data on these variables was not available for every model for WB2. We report results on each variable for the models where it was available. These experiments find generally the same results as those on T850 and Z500: systemic bias in model performance across models that increases with lead time.

## E.1    PRECIPITATION

**General fairness.** We look at total precipitation over 6 hour (P6) and 24 hour (P24) periods for GraphCast and FuXi. Greatest absolute difference (Figure 9) and variance (Figure 10) in RMSE are plotted. Except on the landcover attribute at high lead times, GraphCast is more fair than Fuxi. It is interesting that GraphCast is generally fairer than FuXi at high lead times, as FuXi outperforms GraphCast on traditional, globally-averaged performance in those cases (Figure 11).

## E.2    2M TEMPERATURE AND WIND SPEED

**General fairness.** We look at air temperature at 2m (T2m) and wind speed at 10m (U10m and V10m) on GraphCast, Pangu-Weather, and FuXi, the only models that had data for these variables.

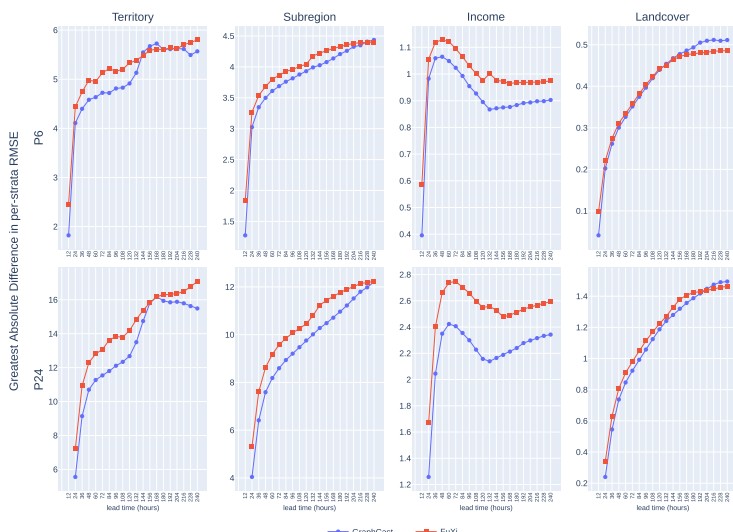

Figure 9: Greatest absolute difference of any two per-strata RMSE for each attribute when predicting precipitation for 6 hours (P6) and 24 hours (P24) at different lead times. Lower difference is more fair.

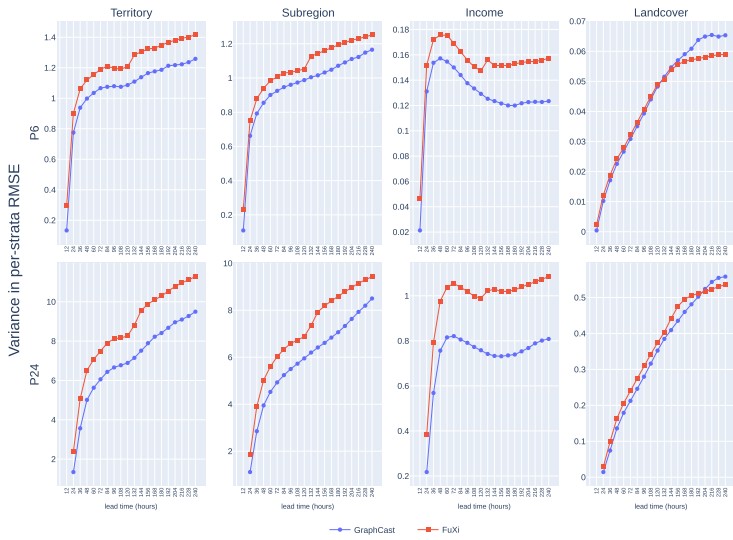

Figure 10: Variance in per-strata RMSE for each attribute when predicting precipitation for 6 hours (P6) and 24 hours (P24) at different lead times. Lower variance is more fair.

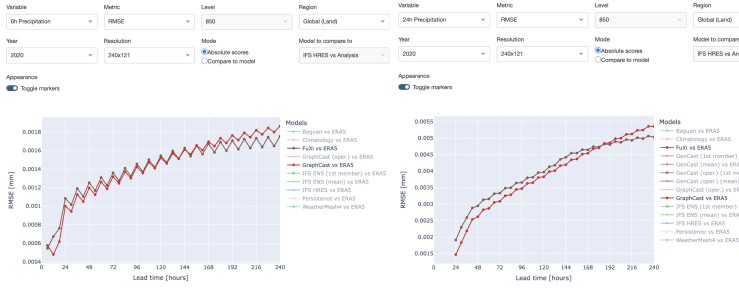

Figure 11: Performance of GraphCast and FuXi on P6 and P24 in terms of globally-averaged RMSE. Lower is better. Screenshot taken from Google's WeatherBench dashboard.

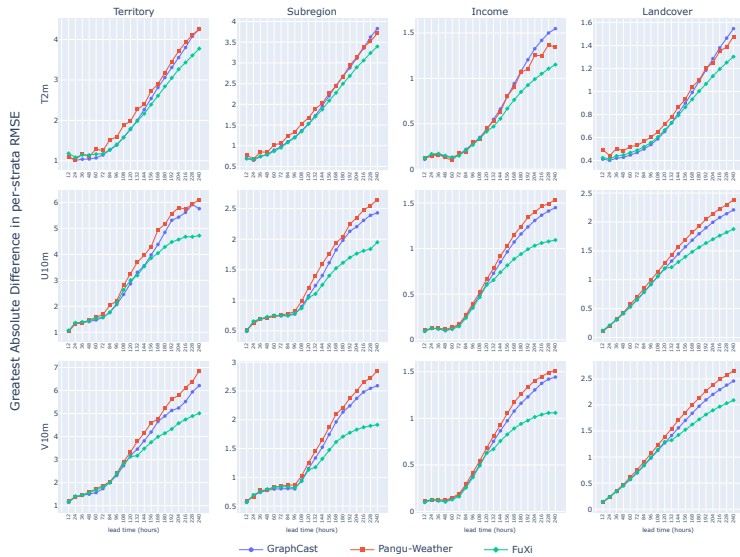

Figure 12: Greatest absolute difference of any two per-strata RMSE for each attribute on 2m temperature (T2m) and the U and V components of wind speed at 10m (U10m and V10m, respectively). Lower difference is more fair.

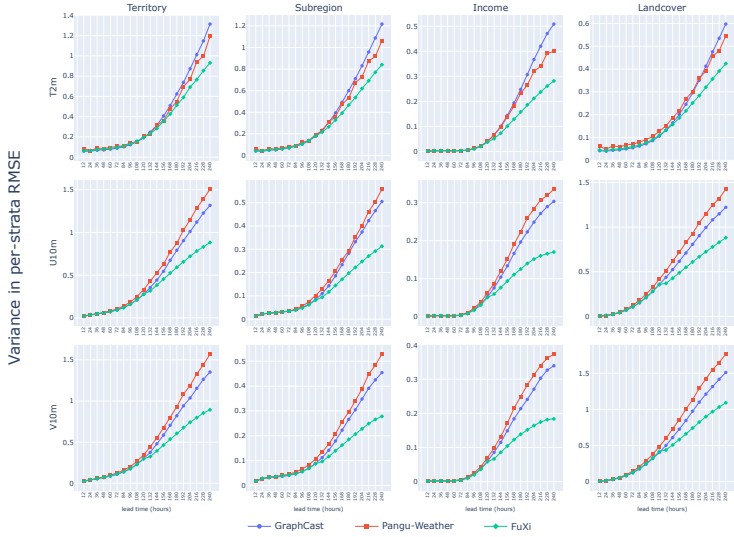

Figure 13: Variance in per-strata RMSE for each attribute on 2m temperature (T2m) and the U and V components of wind speed at 10m (U10m and V10m, respectively). Lower variance is more fair.

Greatest absolute difference (Figure 12) and variance (Figure 13) in RMSE are plotted. For high lead times, FuXi is generally the most fair. At low lead times on 2m temperature, GraphCast is the most fair—an interesting result as it becomes the least fair at high lead times. At low lead times on the wind speed variables, GraphCast and Pangu-Weather alternate being the most fair. At high lead times, Pangu-Weather is the least fair in predicting 10m wind speeds. As with every other variable assessed, unfairness and the inter-model disparities in unfairness grow as lead time grows.

## F  Supplemental figures

Figure 14 depicts the same results as the income attribute experiment in subsection 4.4 while zoomed in on low lead times. It is seen that at low lead times, all models perform worst in lower income countries.

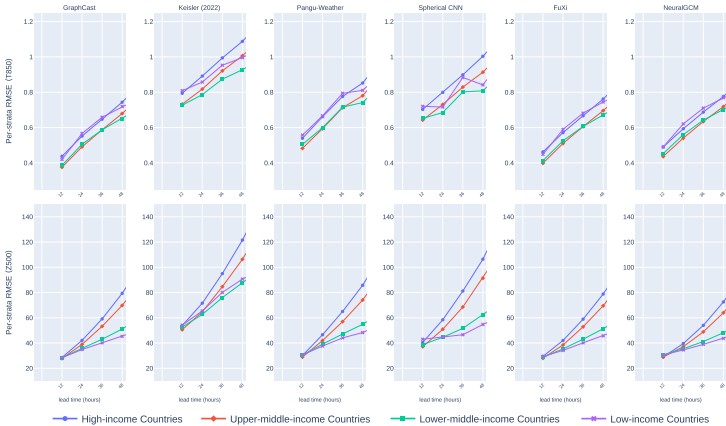

Figure 14: Per-strata RMSE for the income attribute of each model for the first 48 hours of lead time.