# OpenReview forum: "SAFE: Benchmarking AI Weather Prediction Fairness with Stratified Assessments of Forecasts over Earth"
_ICLR.cc/2026/Conference — Submitted to ICLR 2026_

### Official Review · Reviewer_6rWx · 2025-10-26

**Soundness:** 3
**Presentation:** 3
**Contribution:** 3
**Rating:** 6
**Confidence:** 4

**Summary:**

In this paper, the authors study the problem of weather prediction. To be specific, they argue that existing evaluation methods / benchmarks ignore many biases present in prediction models and to address this issue, they introduce SAFE, a fairness evaluation framework for AI-based weather prediction models.

**Strengths:**

+ The authors highlight a novel problem in AI-based weather prediction.
+ They introduce a novel benchmark with its accomponying methods for data collection and evaluation.

**Weaknesses:**

Weaknesses:

1. The paper should introduce a list of concrete desiderata expected from such a benchmark and provide a comparison table with respect to the existing benchmarks. This comparison should also consider benchmark size in terms of samples & locations.

2. Introduction should introduce the research gaps and describe how the paper addresses those gaps.

3. I am not sure ICLR is a good venue for such a paper as I don't think that the weather prediction community is following ICLR.

Minor comments:
- Lines 46-47: "Root mean square error (RMSE) is the preeminent metric used in assessing the quality of AIWP models (Radford et al., 2025; Rasp et al., 2020). The general form of RMSE is shown in Equation 1," => This paragraph is incomplete and a bit out of place here.

- Please use the following guide for writing equations:
https://wp.optics.arizona.edu/kupinski/wp-content/uploads/sites/91/2023/05/MerminEquations.pdf

- Eq 1: Please introduce the symbols.

- "high resolution events" => What is a high resolution event? Did you mean high frequency?

- "see: section 2" => "see Section 2".

- "package 1" => Footnotes should be placed without spaces.

- "In calculating the loss function for training it is common to weight the (squared if L2) difference in variable prediction and ground truth by the area of the gridpoint cell the forecast was made at before averaging." => Please cite.

- "Q3 + 1.5 ∗ IQR)" => "*" denotes convolution, not multiplication. Use \times for multiplication.

**Questions:**

Please see Weaknesses.

---

> ### Author Response · Authors · 2025-12-03
>
> Thank you to the reviewer for their thoughtful feedback and keen typographical corrections. Such a close read is greatly appreciated.
>
> Rebuttal to weakness:
>
> 1. We describe fairness metrics desiderata in subsection 3.2.3. We add some comparison with existing benchmarks in Appendix E.
> 2. We have made the Introduction clearer in terms of our contributions.
> 3. The mission of ICLR 2026 includes publishing work on “societal considerations” and “climate,” the main categories of our paper. At ICLR 2025, myriad papers dealt with weather forecasting [[1](https://iclr.cc/virtual/2025/poster/28928), [2](https://iclr.cc/virtual/2025/poster/28660), [3](https://iclr.cc/virtual/2025/poster/30500), [4](https://iclr.cc/virtual/2025/poster/30964)]. There were two workshops that highlighted forecasting and climate work: [Tackling Climate Change with Machine Learning](https://iclr.cc/virtual/2025/workshop/23966) and [XAI4Science](https://iclr.cc/virtual/2025/workshop/23980). In any case, the work is targeted toward AI/ML researchers that are developing the forecasting models, to make them aware of the biases they produce, as much as the model users. We believe it is important to highlight these issues in core AI/ML venues, as fairness and social impact is far too often ignored by the field.
>
> In response to your minor comments:
>
> - We inserted text at lines 54-57 that had been accidentally omitted. It completes the paragraph and introduces the mathematical notation used in Equation 1 and throughout the rest of the paper.
> - We adjusted Equation 1 to follow the suggested equation writing guidelines, with the main change being treating Equation 1 as prose integrated in the paragraph.
> - Explanation of variables included in lines 54-57.
> - We have removed the portion of Section 1 discussing “high resolution events”. This language reflected an older version of the manuscript, and is no longer germane to the novel contributions of our final work.
> - This text was removed as redundant.
> - Whitespace before footnote removed.
> - Citations added in the sentence following the highlighted on. Specifically, we mention how all six models we evaluate use this method, as well as other models in the literature including ones that are new as of this year.
> - We change our method of outlier detection from Tukey fences to local outlier factor (LOF) because of the nature of the distribution of RMSEs. Experimental control for outliers identified with this new method has been added in Appendix C. Because we no longer utilize Tukey fences, the misformatted formula has been removed. However, we would like to express our deep appreciation to the reviewer for catching this symbol typo.

---

### Official Review · Reviewer_mBAM · 2025-10-31

**Soundness:** 2
**Presentation:** 3
**Contribution:** 2
**Rating:** 2
**Confidence:** 4

**Summary:**

The paper aims to provide a benchmarking framework for evaluating fairness in AI weather prediction models by detailed, stratified error analysis across geographic, economic, and environmental groups rather than just global averages.​  It measures model performance (mainly RMSE) for each "stratum," e.g., country, region, income group, or land vs. water, using accurate, area-weighted calculations. It introduces per-stratum disparity and variance metrics to quantify fairness. SAFE is applied to several AI weather prediction models, demonstrating disparities in accuracy across all attributes and lead times.

While the paper addresses an important issue, I feel that the evaluation is quite limited and that the work should be extended to be a more comprehensive benchmark, encompassing further strata, and probabilistic models.  Moreover, I would argue that such works primarily concerning Atmospheric and Climate science might be better suited to domain journals, and their target audience.

**Strengths:**

- Highlights hidden biases by moving beyond crude spatial averages.​

- Provides fine-grained, interpretable error benchmarks for specific regions, income groups, and land types.​

- Metrics are grounded in fairness literature and extensible.​

- More accurate latitude weighting that accounts for Earth's oblate shape.​

 - Open-source and reproducible.​

- Shows systemic bias exists across models and geographies, which is critical for real-world deployment.​

**Weaknesses:**

- It is well-understood in the literature that RMSE is missing the mark when it comes to these models as it encourages unphysical predictions that blur out solutions. As such, the literature has moved to assess performance by measuring CRPS, spectral fidelity and llocalized scores. The paper does not address or acknowledge any of this and crucially, does not provide any hints on how this could be appl;ied to the more modern, probabilistic models.

- Some claims are made without being backed up. For instance, it is well-known that the Earth is oblate but I doubt that this has a serious effect on either scoring or even on the absolute RMSE values. For instance, apart from FourCastNet3, most ML weather models do not even consider quadrature weights to compute spatial averages on the sphere, without apparent downsides.It would be good to back this up with numerical proof that such changes do have a measurable effect.

- Similar to the previous point, it would be appropriate to demonstrate the usefulness of SAFE by showing how it affects the ranking of models as compared to globally averaged RMSE.

- Often the choice of metric will even lead to different rankings, so one strata to consider would also be performance on extreme events such as tropical cyclones or floods. In this setting the probabilistic nature would be especially important and I expect models that perform well RMSE-wise to perform poorly in this setting.

- Current attributes are limited and fairness metrics are basic: work is needed to incorporate more nuanced strata (like coastlines, islands, or population density).​

**Questions:**

- How much of a difference does it make to consider the oblate geometry, really?
- Why not include probabilistic models and more importantly, consider extreme events? There are several datasets providing and extreme events record.

---

> ### Author Response · Authors · 2025-12-03
>
> We thank the reviewer for their response. We view our critical target audience as the AI/ML researchers that are developing forecasting models. We seek to make them aware of the biases they produce. As such, it is necessary to highlight these issues in core AI/ML venues, as fairness and social impact is far too often ignored by the field.
>
> Rebuttal on weaknesses:
>
> - The reviewer’s characterization that RMSE is no longer used in the field is wholly inaccurate (e.g., Section 5 of [[1](https://arxiv.org/pdf/2507.12144)] published in July 2025). This is evident from our Introduction. To address the second part of this comment: a common method in the field for assessing skill of probabilistic models is to take an ensemble and apply RMSE to the average prediction. Thus, our approach is directly applicable to probabilistic models. Lastly, it the modular design of the open source SAFE package we introduce makes it possible to incorporate future metrics as the reviewer desires.
> - Can the reviewer clarify if by “quadrature weights” they mean (1) precise calculation of grid cell area that accounts for Earth’s oblate shape, or (2) any method that calculates grid cell area with a simplified spherical model. If (2), then the reviewer’s claim that most ML methods do not incorporate area weights is inaccurate. Virtually all ML weather models incorporate area weights. We have added text to subsubsection 3.2.2 to demonstrate this:
>
> >  In calculating the loss function for training it is common to weight the (squared if L2) difference in variable prediction and ground truth by the area of the grid cell the forecast was made at before averaging. This is the case with all six models we assess: GraphCast (Lam et al., 2023) Supp. Mat. Eq. 19, Keisler (Keisler, 2022) section 3.3.3, Pangu-Weather (Bi et al., 2022) Eq. 2, Spherical CNN—which refers to Rasp & Thuerey (2021) for training details, FuXi (Chen et al., 2023) Eq. 2, and NeuralGCM (Kochkov et al., 2024) Supp. Mat. G.4; as well as in other state-of-the-art models including but not limited to GenCast (Price et al., 2023) Supp. Mat. D.4, FGN (Alet et al., 2025) Eq. 5, and FourCastNet 3 (Bonev et al., 2025) Eq. 50.
>
> - We add some discussion of this in Appendix E.
> - Again, RMSE is a widely used statistic for measuring the skill of probabilistic models.
> - We agree that adding more attributes is an important line of future work. However, the only attribute for stratification commonly used in the field already is subregion, as we note in subsection 2.1. Even then, the subregions are defined poorly and main results are not stratified even on these. Thus, it is a significant contribution for us to so thoroughly improve subregion stratification and introduce three altogether new ones.
>
> Response to questions:
>
> - As we already mentioned in subsection 3.2.2, spherical models can overrepresent the area of grid cells by 504%. This matters as area weighting is an approach used in virtually all AI weather prediction work, also noted in the same subsection.
> - We believe it is important to start with the most fundamental use case of AI weather prediction models, hence our focus on standard use cases. The models we already include are being used for genuine weather prediction, such as [ECMWF’s use of GraphCast](https://charts.ecmwf.int/products/graphcast_medium-mslp-wind850?base_time=202512031200&projection=opencharts_europe&valid_time=202512031200). We agree with the reviewer that including more types of models and different datasets is a good direction for future research.

---

### Official Review · Reviewer_6egV · 2025-11-01

**Soundness:** 3
**Presentation:** 3
**Contribution:** 3
**Rating:** 6
**Confidence:** 3

**Summary:**

The paper introduces an open-source framework for evaluating the fairness of ML weather prediction models. Instead of relying on globally averaged metrics like RMSE, SAFE assesses model performance across geographic and socioeconomic strata—such as country, income level, and landcover—to reveal spatial disparities. The paper thus establishes the first benchmark for geographically stratified fairness in weather forecasting.

**Strengths:**

The paper is among the first to demonstrate a solution for assessing fairness in ML weather prediction, a dimension that is often ignored in existing work that typically focuses only on global accuracy.
The authors nicely expose systemic geographic and economic biases in widely used ML weather models.

**Weaknesses:**

- While SAFE introduces valuable stratifications, it currently includes only four attributes (territory, subregion, income, landcover), omitting potentially important ones like population density, climate zone, or infrastructure exposure.
- The experiments use only two atmospheric variables (T850 and Z500), which may limit generalizability to other variables, resolutions, or forecasting tasks.
- The proposed fairness measures (RMSE variance and maximum difference) are basic statistical descriptors and may not capture more nuanced or causal forms of bias.

**Questions:**

What factors, e.g., data distribution, model architecture, or physical representation, cause the observed geographic and socioeconomic disparities in forecast accuracy, and how might SAFE be used to mitigate them during model training?

---

> ### Author Response · Authors · 2025-12-03
>
> Thank you to the referee for their considerate review. We appreciate that they acknowledge how novel our work is applying ML fairness to AI/ML weather prediction.
>
> Rebuttal on weaknesses:
>
> - Including more attributes is indeed a good future direction for the package, and we acknowledge this in section 5. However, the only attribute for stratification commonly used in the field already is subregion, as we note in subsection 2.1. Even then, the subregions are defined poorly and main results are not stratified even on these. Thus, it is a significant contribution for us to so thoroughly improve subregion stratification and introduce three altogether new ones.
> - We add experiments on 5 additional variables in Appendix E: precipitation (6hr and 24hr), 2m temperature, and wind speed at 10m (U and V components). The results on these variables find the same conclusion as with T850 and Z500: there are systemic biases across all models, on all attributes, for all variables, that increase with lead time. To clarify, SAFE as a package can be used flexibly by the end-user on any dataset and any susbset of variables. The 7 variables we look at are the most commonly used in the AI weather prediction literature for measuring model skill, with particular focus in the field on T850 and Z500 (as we explain in subsection 4.2).
> - The metrics we consider are among the state of the art in the machine learning fairness field. We justify this with an expanded literature review in subsection 2.2. In fact, our aim to measure fairness on continuous regression across more than two strata is a nearly entirely novel contribution in the field.
>
> Response to questions:
>
> - We agree that understanding the causes and remedies for the disparities is important follow up work. We acknowledge this in section 5, but this work is out of scope for this paper. The creation of the SAFE package opens up the door to those future works. And our discovery that systemic biases exist is a significant contribution to the field of AI weather and climate prediction, one hitherto not reported on, and that is the aim of this paper. These results upend the fundamental approach of spatial averaging in loss functions and evaluation.

---

### Official Review · Reviewer_6Syp · 2025-11-01

**Soundness:** 3
**Presentation:** 3
**Contribution:** 3
**Rating:** 6
**Confidence:** 4

**Summary:**

This paper introduces SAFE, a Python-based open-source toolkit for evaluating fairness in AI-based weather prediction (AIWP) models through stratified performance assessment. Unlike standard evaluation protocols that report globally-averaged RMSE or ACC, SAFE enables model performance analysis across territory, global subregion, income level, and landcover (land/water). The authors evaluate six leading AIWP models (GraphCast, FuXi, Keisler, Pangu-Weather, Spherical CNN, and NeuralGCM) across multiple lead times (12h to 240h), over two key variables (T850 and Z500), using ERA5 data from 2020, and report disparities in per-strata RMSE. The paper further defines two new fairness metrics—maximum RMSE disparity and variance across strata—to benchmark inter-model fairness.

**Strengths:**

* Evaluation across 6 major models, 4 fairness attributes, 20 time steps, and 2 variables.
* The SAFE package is open-source and built on public datasets, facilitating reproducibility.
* The use of pygeoboundaries_geolab together with an oblate-spheroid area correction improves the accuracy of area weighting and geographic stratification.

**Weaknesses:**

* While novel in this context, both max RMSE difference and RMSE variance are basic summary statistics. The paper acknowledges that fairness metrics from ML are typically binary or categorical, but does not attempt any adaptation of more sophisticated metrics.

* The authors do not consult with any domain experts (e.g., meteorologists or users in Global South) to validate whether the stratified results align with known experience.

* The study focuses only on T850 and Z500. However, precipitation and extreme events, which are crucial for human safety and fairness (e.g., flood warnings), are omitted.

* Although the authors motivate the work with real-world consequences (e.g., extreme heat mortality), they do not quantify the societal impact of the observed disparities.

**Questions:**

See weaknesses

---

> ### Author Response · Authors · 2025-12-03
>
> We thank the reviewer for their feedback! We find your summary particularly insightful towards understanding how are paper will be received and understood.
>
> Response to weakness:
>
> - As you correctly note, existing metrics almost exclusively operate on binary outcomes. The sole existing work we have found for continuous prediction spaces used a metric that only works for 2 strata. As the reviewer desires, we have adapted this method and generalized it to our >2 strata context through use of maximum difference and variance metrics. Though the reviewer may not realize it, this does in fact at the cutting edge of fairness metrics that have been developed for regression problems. We have added subsection 2.2 to provide a literature review that substantiates this claim.
> - We have reached out to domain experts, but have not received enough feedback to incorporate anything meaningfully into the paper.
> - We add experiments on precipitation (both 6hr and 24hr cumulative variables), as well as 2m temperature, U component of 10m wind speed, and V component of 10m wind speed in Appendix E. T850 and Z500 are still the most commonly used variables for reporting model skill (as we note in the paper), but these 5 additional variables we have added are important for reasons the reviewer notes. The findings on these variables are nearly identical to our previously reported results on T850 and Z500, supporting our claims of systemic bias.
> - Work in this general direction is important, but its pursuit is downstream of our groundbreaking paper. That is, we for the first time make the field aware that this is a question to ask. Without our work, the field is unaware that these disparities even exist, much less that there are downstream impacts or harms occurring.

---

### Author Response · Authors · 2025-12-03
**Details on Paper Revisions**

Thank you to all of the referees for their thoughtful reviews. We have uploaded a new version of our paper to incorporate referee feedback. In this comment we detail updates which address common critiques.

### Major updates since submission:

- **Updated title/abstract.** Updated the title to reflect the fact that the main contribution is our novel approach to evaluation through stratification and fairness metrics, and that doing so reveals hitherto undetected biases in models. The provided benchmark is a downstream application of this new methodology that compares relative amounts of bias.
- **New experiments at strata level.** We add experiments in subsection 4.4: “Income attribute” and “Landcover attribute.” In these sections we dig deeper, analyzing the per-strata RMSE rather than just the summary fairness metrics. This is tractable with these two attributes because of their small strata counts (i.e., it would be unhelpful to have graphs trying to simultaneously visualize the dozens and hundreds of strata within the region/territory attributes). Of note, we find that at low lead times (i.e., predictions for weather within the next day or two), most models perform worst in low-income countries. This is an impactful finding with serious implications for ethical considerations in model deployment.
- **More climatic variables.** We redo our experiments for additional climatic variables: precipitation (both 6hr and 24hr cumulative), surface temperature, and wind speed (10m U and V components). These variables represent the second tier in popularity for reporting model performance in the literature after the ones we already use. Results are in the appendix as data on these variables was not available for every model.
- **Control for outliers.** We add a genuine control for outliers in Appendix C, replacing the text at the end of the “General fairness” paragraph of subsection 4.4 that merely characterized outliers. In Appendix C, we remove outlier RMSEs and re-conduct the same experiment: measuring greatest absolute difference and variance in RMSE across the strata. We find that near-equivalent levels of disparities exist, and that the metrics trend the same way across lead time for all attributes. This shows that the results we find are not being driven by outliers, but rather reflect a systemic issue.
- **Grounding in fairness literature.** Added subsection 2.2 to discuss related work in fairness for continuous regression. The findings are the same: there is little to no existing work that addresses fairness in the continuous outcome space. The work that does exist supports our approach of taking the difference in average per-strata (i.e., group) performance, as these works take very similar approaches. However, they often only operate in the two strata setting, so by considering more than two strata at a time, our metrics push the field further. We reiterate that as more fundamental results in ML fairness are published, they can be seamlessly integrated into the SAFE package through our modular design, and that pushing the theoretical field of fairness metrics is not the main contribution (even though we do so), it’s the application of such metrics to discover the biases in weather prediction. Based on our related work findings we are at the cutting edge of state of the art fairness metrics already.

### Minor updates:

- Our statement on not using LLMs in paper writing will be moved to the Acknowledgements section. This is a more appropriate spot than in the technical appendix. The Acknowledgements section is hidden to maintain double-blind review. It is still true that we used no LLMs in writing our paper or responses.
- Appendix B.3 (formerly C.3) incorrectly cited the LandScan dataset as providing landcover strata labels. This has been corrected with a proper citation to the GMT dataset.
- Added discussion of GraphCast and WB2 as examples of previous work in Section 2. Made the contribution of the already-cited NeuralGCM more clear.
- Fixed various spelling and typographical errors.

---

### Meta-Review · Area_Chair_iweP · 2026-01-02

**Summary:**

The paper introduces a benchmarking framework for incorporating fairness in the evaluation of AI weather prediction (AIWP) models. All reviewers acknowledge the importance of such evaluation, but also highlight several issues with the paper.

The method focuses on simple summary statistics (RMSE difference/variance) which might not capture more nuanced notions of fairness in the predictions. The framework should be better anchored among domain experts and focus on relevant use cases that have more directly societal impact (extreme events etc.). Furthermore, the evaluation focuses on deterministic models which are known to have different qualitative behavior than generative models, not least for accurately representing high-frequency spatial variations.

There are also concerns regarding whether or not ICLR is a suitable venue for this line of works. The authors argue that it's important the the AI community is aware of the fairness aspects of the AIWP models developed by this community. This is of course true, but I still believe that there is a mismatch between the objective of this paper and the focus of ICLR, partly because the notion of fairness in weather predictions extends beyond AIWP models.

**Reviewer Concerns:**

Some of the comments have been partially resolved (e.g. including precipitation in the evaluation) but the overall issues as mentioned above remain.

**Reviewer Scores:**

The paper has three 6's, but my feeling is that these scores do not properly reflect the criticism brought forward in the corresponding reviews. The reviewers largely agree on the limitations of the work. I don't that the reviewers would have changed their scores in light of the rebuttal.

---

### Decision · Program_Chairs · 2026-01-26

Reject